# Reproducibility of real-world evidence studies using clinical practice data to inform regulatory and coverage decisions

Shirley V. Wang [1,2] ✉, Sushama Kattinakere Sreedhara[1],
Sebastian Schneeweiss[1,2] & REPEAT Initiative*

Studies that generate real-world evidence on the effects of medical products through analysis of digital data collected in clinical practice provide key insights for regulators, payers, and other healthcare decision-makers. Ensuring reproducibility of such findings is fundamental to effective evidence-based decision-making. We reproduce results for 150 studies published in peer-reviewed journals using the same healthcare databases as original investigators and evaluate the completeness of reporting for 250. Original and reproduction effect sizes were positively correlated (Pearson's correlation = 0.85), a strong relationship with some room for improvement. The median and interquartile range for the relative magnitude of effect (e.g., hazard ratio$_{original}$/hazard ratio$_{reproduction}$) is 1.0 [0.9, 1.1], range [0.3, 2.1]. While the majority of results are closely reproduced, a subset are not. The latter can be explained by incomplete reporting and updated data. Greater methodological transparency aligned with new guidance may further improve reproducibility and validity assessment, thus facilitating evidence-based decision-making. Study registration number: EUPAS19636.

The vast amount of digital information produced in healthcare is increasingly transformed into real-world evidence (RWE) on the safety and effectiveness of medical products in clinical practice, and playing a critical role in decision-making for regulators, payers and physicians[1–4]. Advances in the understanding of valid study design and analysis of longitudinal healthcare data for causal inference[5–8] have made it possible for database studies to both reproduce[9–12] and predict[13–16] results from randomized clinical trials (RCT), as well as provide evidence that changed clinical practice in situations where trials were not feasible[17–19].

The need for timely, high-quality clinical evidence from data generated in clinical practice has become urgent during the COVID-19 pandemic[20]. However, an influx of high-profile RWE studies with methodological shortcomings, some of which have been published and retracted[21,22], has contributed to negative generalizations about the credibility of RWE rather than focusing on distinguishing valid and

robust methodology from that which is less reliable[23,24]. This, coupled with the lack of clarity in reporting on study implementation, which has been one of the most frequently mentioned barriers for healthcare decision-makers against use of RWE[25], has reduced stakeholders' confidence in study findings[26,27].

Actionable scientific evidence in medicine should be internally valid, reproducible, and replicable. Reproducibility is the ability to obtain the same results when reanalyzing the original data, following the original analysis strategy. Replicability is the ability to confirm findings in different data and populations. In principle, all reported findings should be perfectly reproducible[28]. A finding can be reproducible but invalid because of problems in sampling, study design, measurement, or statistical inference. But, if a finding is not reproducible, there is little basis for evaluating its validity or replicability. Therefore, achieving reproducibility is a fundamental step for research credibility.

[1]Division of Pharmacoepidemiology and Pharmacoeconomics, Brigham and Women's Hospital, Boston, MA, USA. [2]Department of Medicine, Harvard Medical School, Boston, MA, USA. *A list of authors and their affiliations appears at the end of the paper. ✉e-mail: swang1@bwh.harvard.edu

Computational reproducibility needs only two ingredients—shared data and analysis programming code. While very useful, it can be difficult to ascertain intended scientific decisions from programming code alone, much less evaluate the validity of those decisions. This is because extracting an appropriately temporally anchored analytic study cohort from longitudinal source data tables can involve thousands of lines of code[28], making it challenging for even experienced programmers and data scientists to parse and identify the many relevant scientific design and analytic decisions that were implemented to minimize bias.

While computational reproducibility will allow reproduction of the same exact result to the nth decimal place, independent reproducibility focuses on effective communication of critical design and analytic choices (with the potential to affect the validity, relevance or interpretation of results). This clarity is necessary for assessment of potential sources of bias (e.g., confounding, misclassification, selection bias) as well as to facilitate replication of findings with data that is stored in a different data model. Thus, independent reproducibility is critically important because it reflects the clarity of communication of key study design and analytic parameters, regardless of whether shared data and analysis code are available.

Previous pilot work evaluated the independent reproducibility of a small, non-systematic sample of RWE studies[29]. The challenges with reproducing the studies in the convenience sample motivated this project, which involved the systematic identification of a large random sample of RWE studies fitting pre-specified parameters. Similar to large scale evaluation of research reproducibility in other disciplines[30–32], we sought to (1) describe the frequency of reporting key parameters about data transformations, study design choices and statistical analysis needed to ensure independent reproducibility[33] and (2) evaluate the independent reproducibility of results from 150 published RWE studies using the same healthcare databases and applying the same reported methods as original authors. For each study, we independently attempted to reproduce the study population and primary outcome findings by conducting the analyses as described in the original papers, appendices and other public material, and by making informed assumptions on study parameters that were not reported, while being blinded to the study findings.

In this systematic, descriptive review, we show strong correlation in results between the original and reproduced RWE studies and provide insights on how to further improve reproducibility for the subset where results diverged.

## Results
### Study sample
The sampled studies are described in the Supplementary materials and methods.

### Clarity of reporting
We evaluated clarity of reporting for 250 identified studies based on key study parameters identified in a reporting consensus document[33]. An attrition table or flow diagram showing counts as inclusion-exclusion criteria were applied was provided in 54% of the sampled studies. Design diagrams to communicate key aspects of study design were provided in 8%. The proportion of studies clearly reporting study implementation parameters varied depending on the parameter (Supplementary Table 1). For example, the criterion to define the cohort entry date (e.g., the index date for subjects entering the study population), was reported in 89% of the sample. In contrast, for studies involving measurement of the duration of exposure, the algorithms used to operationally define duration frequently were not provided (≤55% reported). Such algorithms could include those used to calculate duration in situations when the data indicate that there were early

refills (resulting in overlapping days supply), or algorithms used to extend the definition of exposure beyond days supply dispensed (to allow for non-adherence or capture the hypothesized window of biologic effect). Out of 6 categories for comparative studies (each category combining multiple study parameters used to define index date, inclusion-exclusion criteria, exposure, outcome, follow-up, covariates), the median and interquartile range for the number of categories where the reproduction team made assumptions was 4 [3, 5]. Out of 5 categories for descriptive studies (where no specific exposure was studied) the median and interquartile range for the number of categories where the reproduction team made assumptions was 3 [2, 4] (Supplementary Table 1). Only 3 out of 250 studies did not require an assumption in any of these categories. Analytic code in the form of macros, other open-source code, or specific procedures was referenced in 7% of reproduced studies. However, the exact software version or selected options that were used to run the code for these studies were only partially provided. Operational algorithms to measure outcomes, including clinical codes (e.g., International Classifications of Diseases), care setting (e.g., inpatient versus outpatient) and diagnosis position (e.g., primary versus secondary), were more frequently provided than algorithms for inclusion-exclusion criteria and covariates across the sampled studies (Supplementary Table 2).

### Reproduction of population size
The median and interquartile range (IQR) for the relative sample size (original/reproduction) of study reproductions compared to the original studies was 0.9 [0.7, 1.3] and 0.9 [0.7, 1.0] for comparative and descriptive studies respectively (Fig. 1). For 21% of reproduced studies, the reproduction study size was less than half or more than 2 times the original.

### Anecdote 1—Difficulty reproducing study size: Ambiguous temporality around study entry date
A study conducted in patients with chronic obstructive pulmonary disease (COPD) had an inclusion criterion that required more than one test result confirming COPD, but was unclear about when those disease-confirming tests were required to be recorded before and/or after the COPD diagnosis that defined the study entry date[34]. The Read codes (standard clinical coding system used in primary care in the United Kingdom) used to define COPD diagnosis were not specified, therefore the reproduction team assumed that a Read code algorithm from the National Health Service Quality and Outcomes Framework was used. These factors, plus other ambiguities in the inclusion-exclusion criteria, such as whether they were applied before or after selection of the study entry date contributed to a 26% difference in sample size between the original study cohort and the reproduction cohort.

### Reproduction of baseline characteristics of the study population
The median and IQR for the difference in prevalence (original−reproduction) of baseline characteristics reported in the original publications compared to the reproductions was 0.0% [−1.7%, 2.6%] (Fig. 2). The absolute difference between the original publication reported and the reproduced baseline characteristic prevalence was >10% for 17% of the reproduced characteristics across all studies. Whether the individual codes for a particular covariate algorithm were provided did not explain the ability to reproduce a covariate prevalence (difference <10%), highlighting the influence that lack of clarity on other aspects of covariate measurement (e.g., care setting and assessment window), and ambiguity in the algorithms used for inclusion-exclusion criteria can have on the reproducibility of baseline study characteristics. The distribution of differences was similar for descriptive versus comparative study types as well as for studies where

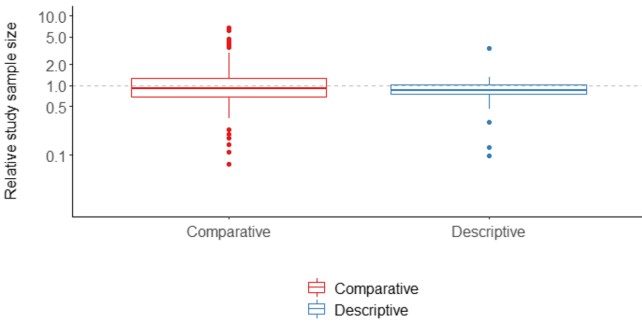

**Fig. 1 | Relative magnitude of sample size (original/reproduction).** Study sizes summed across compared exposure groups for both the original and replication cohorts. Y-axis ticks on log scale, axis labels reflect actual value. Dashed horizontal gray line at 1.0 reflects equal sample size in original and reproduction cohorts. Boxplot elements: center line, median; box limits, upper and lower quartiles; whiskers, 1.5x interquartile range; points, outliers. $N = 118$ comparative, $N = 32$ descriptive studies. Source data are provided as a Source Data file.

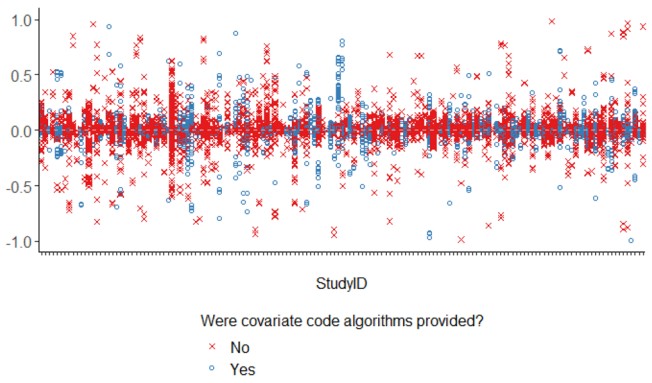

**Fig. 2 | Difference in prevalence of baseline characteristics (original–reproduction).** Each column represents covariates from a different study. Each point represents the difference in corresponding cell values from tables describing characteristics of the cohort in the original paper and the reproduction. Dashed horizontal gray line at 0.0 reflects no difference in prevalence of baseline characteristic between original and reproduction cohorts. Source data are provided as a Source Data file.

there were small, medium or large differences sample size between the original and the reproduction (Supplementary Fig. 2).

## Anecdote 2—Difficulty reproducing baseline characteristics: Missing details

In a study conducted in patients with breast cancer, the authors reported using a modified Charlson comorbidity score as a baseline characteristic, measured in the 3 months before the cohort entry date[35]. One of the components of the published score is tumor/malignancy, with a weight of 2 for patients with the characteristic. The score is the sum of the weights for all components. The original study reported the proportion of patients with a score of 0 vs 1+, with 97% of patients reported to have a comorbidity score of 0. In our study reproduction, only 12% of included patients had a score of 0 (these were patients who met the breast cancer criteria on the cohort entry date but had no cancer codes in the 3 months prior). It was not clear to the reproduction team what modifications were made to the comorbidity score. In the absence of that information, and blind to the results, the reproduction team assumed that a standard operational definition was used. Speculating that the authors could have modified the score to remove the weights for tumor/malignancy, in post hoc exploration, we observed that 74% of patients in our reproduced cohort had a score of 0 if the tumor/malignancy component was removed.

## Reproduction of outcome risks and rates

The median and IQR for the difference in outcome risks and rates (original−reproduction) reported in the original publications compared to the reproductions were 0.0% [−1.5%, 2.0%] and −0.0 [−0.39, 0.45] per 100 person-years (Fig. 3) (stratified by exposure in comparative studies, overall in descriptive studies). Outcome risks differed by >10 percentage points between the original and the reproduction for 11% of reproduced outcome risks. Similarly, 7% of reproduced outcome rates differed by more than 10 per 100 person-years. The funnel shape of Bland-Altman plots show larger reproduction differences when the averaged risk or rate from the original and reproduction was larger (Supplementary Fig. 4A).

## Anecdote 3—Difficulty reproducing outcome risks and rates: Shifts in underlying longitudinal source data in different data versions

We reproduced a study evaluating the risk of all-cause death with the use of benzodiazepines[36]. The reproduced analytic cohort was larger, older, and sicker than the original analytic cohort. The rate of death was higher by 13–16 per 100 person-years. After investigation, we found that shifts in the underlying data in different data versions played a large role in these differences. Although the data used by the original and reproduction investigative teams were covered by the same data license, the data provider retroactively updated historical years of data in the newer data version used by the reproduction team. This updated data version included a larger sample of patients covered by Medicare Advantage programs (an older and sicker population than the commercially insured) and more death information.

## Reproduction of measures of association (hazard ratio, risk ratio, odds ratio) in comparative studies

The median and IQR for the relative magnitude of measures of association (e.g., hazard ratio_original/hazard ratio_reproduction) was 1.0 [0.9, 1.1] (Fig. 4). The full range in relative magnitude was from 0.3 to 2.1. Different statistics indicated strong correlation between the original and reproduced measures of associations (Fig. 5). The unweighted and inverse variance weighted Pearson's correlation coefficient between the original and reproduced measures of association were 0.85 and 0.79, respectively. The unweighted and inverse variance weighted Spearman's Rank Correlation were 0.82 and 0.87. The intraclass correlation coefficient and 95% Confidence Interval was 0.85 (0.79, 0.89). The distribution in the relative magnitude of measures of association was similar regardless of whether there were small, medium, or large differences in the original and reproduced study sample sizes. (Supplementary Fig. 3). A Bland-Altman plot did not show a clear relationship between effect size and reproducibility (Fig. 6).

The absolute difference in the coefficients for the measure of association produced by the original and reproduction (e.g., | log(hazard ratio_original)−log(hazard ratio_reproduction) |) was ≤0.1 for 36% and ≤0.2 for 62% of reproduced comparative studies. The reproduction estimate was closer to the null than the original publication estimate 52% of the time, suggesting that the original publication measures of association were not systematically larger than the reproductions. The point estimates of the original and reproduction measures of association were on the same side of null 82% of the time. The point estimates for the measures of association and the 95% confidence intervals were on the same side of null 61% of the time. When studies had point estimates on the same side of null, the median and interquartile range for the difference in $p$-values between the original and reproduced studies was 0.00 [−0.05, 0.00]. When the estimates were on opposite sides of null, the average absolute difference in the coefficients was 0.4, compared to 0.2 when the estimates were on the same side of null. There was overlap in the 95% confidence intervals for the publication and the reproduction 86% of the time. In 16% of studies, the $p$-value for

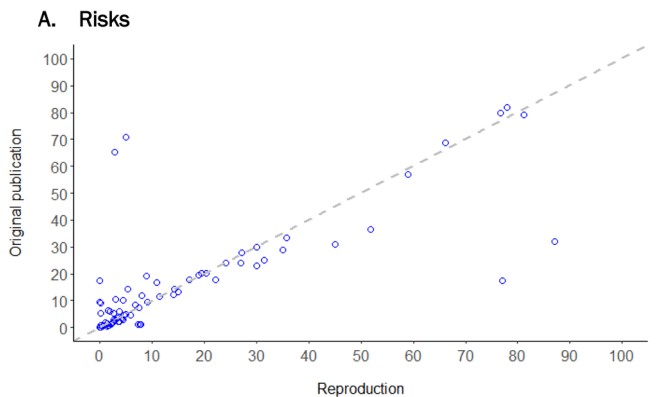

A. Risks

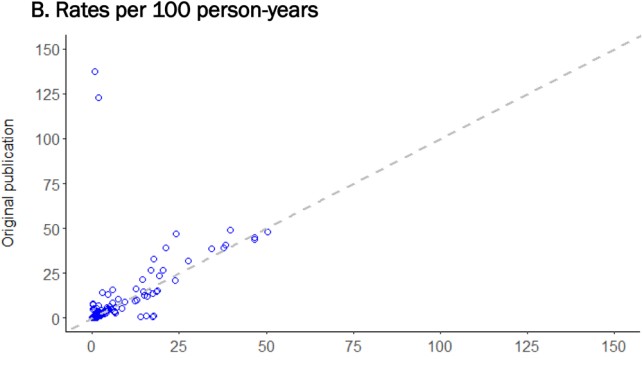

B. Rates per 100 person-years

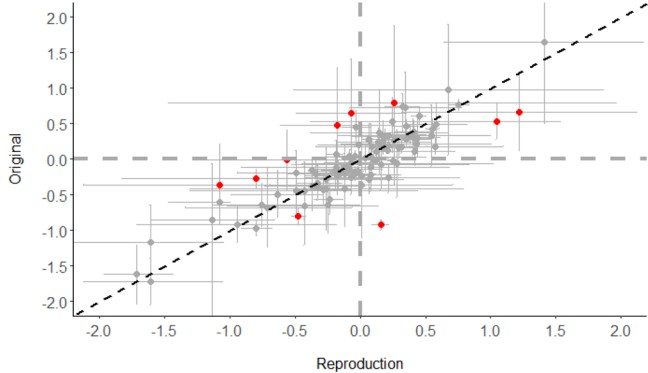

**Fig. 5 | Calibration plot of the logarithms of the hazard ratio, risk ratio, or odds ratio in the reproduction versus the original with correlation coefficient.** Red points reflect the 10 most extreme outliers based on difference between log(effect size) for the original and reproduction. A summary of assumptions for these 10 outliers are in Supplementary Table 4. Error bars reflect the 95% confidence intervals for the original study and the reproduction. Horizontal and vertical dashed gray lines indicate where the original and reproduction effect sizes were null, respectively. The diagonal black dashed line reflects the perfect calibration line where original and reproduced effect sizes would be equal. Source data are provided as a Source Data file.

**Fig. 3 | Calibration in risks or rates of outcome between the original publication versus the reproduction (original − reproduction). A** Risks Dashed diagonal gray line reflects equal outcome risk identified in original and reproduction cohorts. **B** Rates Dashed diagonal gray line reflects equal outcome rates identified in original and reproduction cohorts. Source data are provided as a Source Data file.

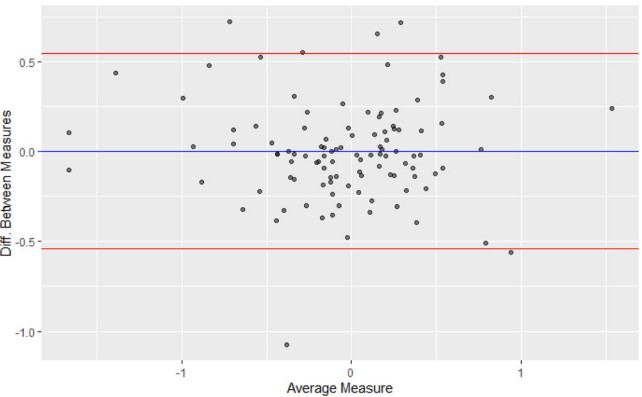

**Fig. 6 | Bland-Altman plot showing the relationship between effect size and reproducibility (difference between original and reproduced coefficients).** The blue horizontal line marks the average difference in log effect size between the original and reproduction. The red horizontal lines mark ±2 times the standard deviation of the difference in log effect size. The circles reflect the values for the difference between the original and reproduction estimates (on the y-axis) and the average of the original and the reproduction estimates (on the x-axis). Source data are provided as a Source Data file.

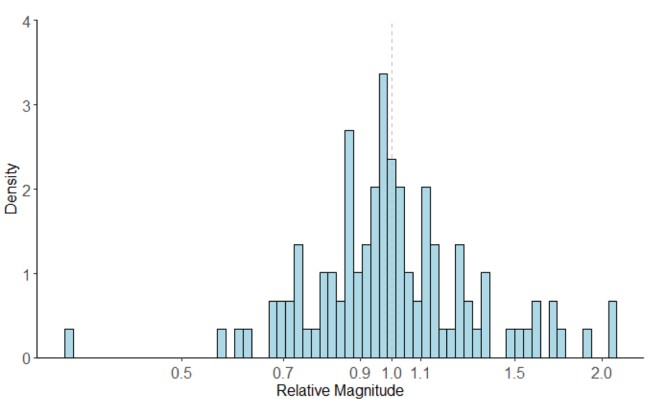

**Fig. 4 | Distribution of relative magnitude of hazard ratio, risk ratio, odds ratio (original/reproduction).** X-axis ticks on log scale, labels reflect relative magnitude. Dashed vertical gray line marks the point at which equal effect sizes were obtained in the original and reproduction. Source data are provided as a Source Data file.

the study reproduction was greater than 0.05 (an arbitrary threshold for failing to reject the null hypothesis) while the *p*-value for the original study was less than or equal to 0.01 (a more stringent threshold for rejecting the null) or vice versa.

**Anecdote 4−Difficulty reproducing measures of association: Error in reporting?**
A study in patients with atrial fibrillation was closely reproduced in terms of study size and baseline cohort characteristics (differences

largely <10%), however the outcome rates were substantially different (e.g., 3-fold higher incidence rate for the reference group in the original compared to the reproduction). It was clearly stated in the manuscript that the outcome algorithm involved relevant clinical codes from an inpatient setting in the *primary* discharge diagnosis position. However, the authors also provided a citation to a validation study for which the outcome algorithm used codes in *any* diagnosis position[37]. In our reproduction attempt, we assumed that the authors used inpatient primary diagnoses as described in the methods section. In post hoc exploration, we used inpatient diagnoses in any position, as described in the cited reference. The point estimate for the original study's hazard ratio was 0.6 times the reproduction estimate when using inpatient primary diagnosis to define the outcome and 1.4 times the reproduction estimate using inpatient diagnoses in any diagnosis position, leaving it unclear whether either corresponded with the parameters used by the original investigators.

### Responsiveness of corresponding authors

Out of 150 authors of the reproduced studies who were contacted, 53% responded, 33% did not respond after 3 attempts at contact, and 13% of the e-mails to corresponding authors were undeliverable. Among 81 responders, 32% ($n = 26$) provided helpful clarification or stated that they were comfortable with our assumptions, 32% ($n = 26$) provided somewhat helpful feedback that did not address all of the assumptions, 27% ($n = 22$) provided responses that did not address the assumptions, and 12% ($n = 6$) declined to discuss their study, citing lack of interest or time.

### Exploring variation in reproducibility of comparative studies

We found little evidence that the average reproducibility (e.g., | $\log(HR_{original}) - \log(HR_{reproduction})$|) for measures of association differed based on clarity of reporting for any single study parameter (Supplementary Table 3). Similarly, the average reproducibility did not meaningfully vary by study characteristics such as whether an author was responsive to our queries, study funding source, study data source, journal impact factor, author citation index, year of publication, or whether the study was originally conducted by members of the same research group that implemented the study reproductions. The differences between the original and reproduced effect sizes were larger on average for studies with larger published effect sizes than studies with smaller published effect sizes ($p < 0.01$). However, this could be an artifact driven by studies with few outcomes that produced highly variable but large effect sizes, or, given the multiple factors explored, a chance finding.

### Reasons for irreproducibility

We summarize assumptions made in the reproduction attempt due to lack of reporting for the 10 studies with the most extreme differences in the coefficients between the original and reproduced measures of association (Supplementary Table 4).

For these 10 outliers, we observed 3 underlying reasons for difficulty reproducing study results.

First, we struggled when there was incomplete information on details of the design and analysis parameters. For some of the more extreme outliers, we particularly noted lack of clarity regarding the temporality of when study parameters were measured relative to the study entry date, as well as absence of detailed algorithms for study parameters.

Second, we observed internally inconsistent information between the text, attrition tables, design diagrams and appendices. Discrepancies forced the reproduction team to make assumptions about which scientific parameters were actually implemented.

Third, shifts in results may have been driven by incomplete information about the source data version. Although we tried to use the same years of data from the same healthcare databases as the original authors, some data sources retroactively update historical years of data, resulting in shifts in the underlying population as well as how certain variables are populated. For example, one of the data sources changed how death was captured in different updates due to privacy concerns. Additionally, reproducibility of studies in some data sources can be further complicated when researchers access is limited to customized subsets rather than full access to the source data. However, information to allow identification of data version is rarely, if ever reported. This may be in part due to the absence of a harmonized system for versioning the semi-fluid data contained in a research healthcare database.

## Discussion

The analysis of massive amounts of individual-level longitudinal healthcare data is becoming more and more common. It can produce evidence on the effectiveness of medical interventions in clinical practice, which in turn informs decision in the healthcare systems that produced the data. Reproducibility of findings derived from such analyses is essential to have confidence in decision making. In 150 RWE studies we showed that the resulting effect sizes were strongly correlated between the original publications and study reproductions in the same data sources and developed insights on where there was room for improvement. While our reproduced estimates were relatively close to the original estimates from the majority of studies, there was a subset of findings that the study team was unable to closely recreate, even though we applied the reported methods to the same healthcare data sources as the original authors.

Our findings are an important calibration metric. This was the largest and most systematic evaluation of reproducibility and reporting-transparency for RWE studies ever conducted. Additional strengths are that the reproduction team was able to acquire access to the same data sources and years of data as reported by the original investigators to independently implement the methods reported in their publications. The reproduction team was blinded to the original study results, so reproductions were implemented based on the methods described, without influence from the published results and the team documented the assumptions and key scientific decisions made for each of the study reproductions. After exhausting publicly provided information to complete the independent study reproductions, the reproduction team approached the original investigative team to clarify ambiguities and omissions in reporting of the original study methods.

Any single metric to characterize reproducibility is imperfect[38]. For example, the proportion of studies where the effect estimate was on the same side of null can mislead in RWE studies with small effect sizes as small implementation differences could conceivably result in enough change to result in an effect estimate on the other side of null in a reproduction attempt. Further, the reproduction effort is focused on US and UK data sources frequently used in research and the generalizability may be limited to well-established and curated research databases that are accessible to independent researchers.

Decision-makers seeking to synthesize RWE to inform their regulatory, policy, or coverage decisions must devote substantial effort to parsing and evaluating the validity of the science behind the results. No specific study parameter stood out as being strongly associated with reproducibility in our univariate descriptive analyses, highlighting the fact that independent reproducibility is multi-factorial. We noted that even studies that were closely reproduced often required considerable discussion within the team, sometimes with many assumptions about the original implementation decisions due to ambiguity in the methods description. The prevalence of studies that could not be closely reproduced speaks to the need for higher levels of transparency and expectations when communicating critical details of RWE study design, analysis, and implementation in protocols, publications and reports. Examining the details of these studies suggested that the divergence was often multi-factorial. Because we observed that the reproduction of most studies required the team to make assumptions on at least one key parameter from a consensus document[33] that outlined elements needed for reproducibility and validity assessment (the basis of our 54-item extraction form), aiming to meet at least this level of detail in reporting on future studies would be a substantial step forward.

Other studies have described the prevalence of issues that bias RWE studies[6,39–43] and we focused on reproducibility without systematic evaluation of the appropriateness of design or analytic choices. While close independent reproducibility of a study is a marker for completeness of reporting on study design and analysis methods, reproducibility is not itself an indicator of high study validity. Indeed, clear reporting in studies with methodological problems enabled our team to closely reproduce results that suffered from the same limitations.

Expanding interest in the use of RWE studies to support health-care decision-making coupled with recognition that ambiguity in reporting limits the utility of such evidence has led to international efforts to develop standards and templates to improve RWE transparency, including clarity in data provenance and processing, exact description of study design choices, details of measurement algorithms and details of statistical analysis[44–51]. These parallel requirements for detailed protocols and statistical analysis plans for randomized clinical trials. For RWE studies, part of transparency on methods to enable independent reproducibility is communicating sufficient detail to allow identification of the relevant data or data version. While many updates to data may have negligible effect on research findings, if there are substantial changes to the contents of the data resources, this may affect the assessment of the fitness of the data resource for the research question.

The need for internationally accepted guidelines to increase the utility of RWE studies for decision-making has led the International Council for Harmonization (ICH) to set as a short-term goal the harmonization of the structure and format of protocols and reporting documents in regulatory submissions[52]. A push to streamline processes that support routine registration of hypothesis-evaluating RWE studies similar to the public registration requirement for trials, is supported by multiple professional organizations[45,53]. Our study fills a critical knowledge gap by providing empirical evidence on the current clarity of RWE study reporting and reproducibility.

Unambiguous communication about the complex data processing, design and analytic choices involved in RWE studies improves understanding of the methodology supporting study findings, the reproducibility of evidence, and the ability to evaluate validity and relevance for healthcare policy decisions. There is always room for improvement and this project has provided insights on how to improve transparency and reproducibility. With coordinated effort from key stakeholders, standards for clear and reproducible RWE studies can be set higher to facilitate efficient evaluation of validity and effective, evidence-based decision-making.

## Methods
### Protocol registration, data, and approvals
The study protocol was reviewed and approved by the Institutional Review Board at Brigham and Women's Hospital. A study protocol was registered at ENCePP prior to selection of the study sample (registration number: EUPAS19636). Each Clinical Practice Research Datalink (CPRD) study reproduction had a study specific protocol approved by an Independent Scientific Advisory Committee (ISAC) prior to implementation of the reproduction. An amendment was filed prior to completing revisions based on reviewer comments.

The use of patient data for this project was authorized by licensed access or data use agreements for 3 administrative healthcare claims and 1 primary care based electronic health record database, each of which is frequently used for research: de-identified Optum Clinformatics claims data, IBM MarketScan Research Database claims data, Medicare fee-for-service claims data, and CPRD electronic health records to conduct study reproductions.

De-identified Optum Clinformatics databases (2004–2017) and IBM MarketScan Research Database (2003–2017) and include national United States claims from employer-sponsored insurance plans for active employees and dependents, early retirees not yet eligible for Medicare, and Medicare-eligible retirees with employer-sponsored non-HMO Medicare Supplementary plans[54,55].

The Medicare fee-for-service data included claims from parts A, B, and D for United States Medicare fee-for-service insurance enrollees over the age of 65. The data included patients with use of anticoagulants (2009-2017), diabetes diagnosis or treatment (2007–2017), and rheumatoid arthritis diagnosis (2006–2017).

Each of these United States administrative claims-based data sources includes diagnosis and procedure codes for billed inpatient and outpatient services, outpatient prescription drug fills, dates of service, and longitudinal health insurance enrollment status.

CPRD data includes de-identified electronic health record data from primary care practices across the UK[56]. The data includes information on patient vitals, signs, symptoms, health related behaviors, diagnoses, immunizations, prescriptions, referrals, and other clinical details. The data also includes information on practices and periods of time during which data collection is considered "up-to-research" standard for practices and patients.

*Scientific Advisory Board*

We engaged a Scientific Advisory Board comprised of international stakeholders invested in understanding the reproducibility of real-world evidence studies. This group included regulators, health technology assessors, payers, academia, industry, contract research organizations, journal editors, professional research societies, and patients. The Scientific Advisory Board convened 5 times over the course of the project, in early meetings focused on developing the overall study design and analysis plan, and in later meetings focused on framing and interpretation of findings.

### Identification of the published studies for reproduction
Our team conducted systematic searches for peer-reviewed database studies published between Jan 1, 2011 and Jun 30, 2017. Eligible studies were required to have been conducted using one of the four data sources that our team had access to, and the years of data used by original investigators were required to match the years of data available to the study team.

We performed a series of systematic searches using Google Scholar. We chose to use Google Scholar because we were interested in studies that were conducted using the specific databases for which we had access. This information is often not provided in the title and abstract of manuscripts. Unlike PubMed and Web of Science, Google Scholar searches the full text when available. The initial search included journals that were (1) highly ranked according to the h-5 index in the Health & Medical Sciences or Epidemiology subcategories in Google Scholar or (2) affiliated with the International Society of Pharmacoepidemiology or the International Society for Pharmacoeconomics and Outcomes Research.

For each journal of interest, we search for results that included the words "cohort" AND "claims," with at least one of the following words contained anywhere within the article: "Optum", "UnitedHealth", "Marketscan," "Truven", "GPRD", "CPRD", "Medicare." The search was limited to results published between Jan 1, 2011–Jun 30, 2017.

In order to boost the sample size of comparative studies, we conducted secondary searches using Google Scholar for comparative cohort studies with restrictions based on publication year and data source. The secondary searches were not restricted by journal.

- Truven MarketScan
  Search google scholar "comparative" AND "cohort" AND "Truven MarketScan"
- Optum Clinformatics
  Search google scholar "comparative" AND "cohort" AND ("Optum Research Database" OR "OptumInsight" OR "Innovus" OR "Optum Lab")
- Medicare Search
  Search google scholar "comparative" AND "cohort" AND "Medicare" AND ("diabetes" OR "rheumatoid arthritis" OR "anticoagulant")
- CPRD
  We reviewed the CPRD bibliography (https://www.cprd.com/bibliography) for publications of comparative studies within the eligible time frame.

The sets of search results were reviewed by a member of the research team to determine whether each article qualified for inclusion in our study. Studies were **excluded** based on:

- *Data source mismatch*: The included studies must have been conducted using a source database for which we had a data use agreement/license. These included Medicare, MarketScan, Optum, CPRD and any combination of these databases if the results were reported separately for each data source. Studies that used other healthcare databases or involved primary data collection (e.g., a randomized clinical trial or animal study), were excluded. If the study used Supplementary linkage of one of these data sources to a database, registry, or electronic health records from a source that was not publicly available, it was excluded.
- *Data source calendar time range*: We excluded published studies that were conducted using years of data that were not included in our license/data use agreement.
- *Full article unavailable*: Studies were excluded if the search result referred to a poster, a conference abstract, pre-print, or members of the review team were unable to access a PDF for a full manuscript.
- *Not a descriptive or comparative safety/effectiveness cohort study*: We required that articles included in the sample were either a descriptive study or had a comparative safety/effectiveness analysis. These were defined as studies that evaluated risk/rate/incidence/prevalence (or comparative risk/rate/incidence/prevalence) of a health outcome for medical health interventions. We excluded articles that did not fall into those categories, including:

  Cost effectiveness analyses.
  Methods papers (e.g., chart review validation, simulation, machine learning algorithm development).
  Review/Meta-Analysis/Letter/Commentary/Editorial/Guidelines.
  Comparison of non-medical interventions (e.g., effect of geography on risk of stroke).

After title and abstract review to identify eligible studies, we ordered studies by publication date, and included the most recent studies that met our criteria. Our target was to evaluate clarity of reporting on 250 studies and to independently reproduce 150 studies. The target was for 80% (120) of the sampled studies for reproduction to focus on a comparative safety or effectiveness question, however, due to an error in categorization early in the reproduction pipeline, only 118 comparative studies were reproduced and an extra 2 descriptive studies were included. The studies were selected from the pool of eligible studies based on reverse date of publication. Over the course of reproduction of some studies, it became apparent that they did not meet the eligibility criteria. When this occurred, we excluded the study and added the next most recently published study that met our criteria. The target sample size for this descriptive characterization of reproducibility was chosen based on estimates for the largest feasible number of studies that could be evaluated within the project time frame.

We reviewed 4,133 abstracts from publications between January 1, 2011 and June 30, 2017 (see supplement materials and Supplementary Fig. 1). Of these, 44.8% were excluded because the data source or date ranges of data available did not match one of the four longitudinal claims or electronic health record databases available for study reproduction, 36.7% were excluded because they were neither descriptive incidence or prevalence studies or comparative studies of a medical intervention (drug, vaccine or device), 12.3% of abstracts were excluded because a full article was not available (e.g., conference abstracts or the article was not in English). This left 6.2% ($N = 258$) of reviewed abstracts eligible for inclusion. Our reproduction sample consisted of 150 of the most recently published papers that met

inclusion criteria, with sampling stratified by comparative versus descriptive studies such that 80% of the sampled studies for reproduction focused on a comparative safety or effectiveness question. Clarity of reporting was evaluated for these 150 studies plus an additional 100 studies, which were included based on recency of publication date. Eight of the selected studies were published by members of the same research department as members of the reproduction team. In these cases, the paper was assigned to research staff and faculty who were not involved with the original publication. The results that are reported reflect independent attempts to reproduce the original publication results, prior to reaching out to the original investigative team.

### Evaluation of reporting clarity of 250 published studies

There were 6 reproduction teams working in parallel on evaluation of reporting clarity and study reproduction. These teams were comprised of at least one faculty member, one masters' or PhD level research scientist, one bachelors level research assistant, and a statistical programmer as needed.

Evaluation of the clarity in reporting of 250 study implementations was based on a standardized extraction form covering 54 items corresponding to elements of a consensus document cataloguing study parameters necessary for reproducibility and validity assessment[33] (Supplementary Data 1). Two members of each reproduction team independently evaluated each paper, with adjudication by a third (senior) team member.

### Empirical reproduction of 150 published studies

For each paper included in the 150 studies sampled for empirical reproduction from the 250 studies described above, the teams focused on reproducing a single study question. One team member redacted pdfs of study materials, including entries of tables, results and discussion sections that provided quantitative information about the cohort characteristics or measure of association. This information was redacted so that the reproduction teams could carry out independent reproductions, without influence from knowledge of the original study result.

The teams reproduced the primary study outcome if it was clearly stated, otherwise, they focused on reproducing the first reported descriptive or comparative result in the abstract. Descriptive results included outcome risk, rate, incidence, or prevalence. Comparative results included effect estimates for comparisons of medical interventions. The reproduction teams implemented design, analysis, and implementation parameters that were reported in the paper, appendices, or citations. However, when parameters were unclear, the team made assumptions. A set of default assumptions was created for study parameters that were frequently ambiguous from publications (Supplementary Data S2). If there was no relevant default assumption and a study parameter was unclear, the assumption was based on the best interpretation of the reproduction team given the context provided in the paper. Each paper had at least 2 research staff and one faculty member involved in the reproduction. The research staff proposed assumptions to make and reviewed these assumptions with the faculty member that they were teamed with. If the team was not able to come to agreement, another faculty member was involved. After discussion, the assumptions made during reproduction were documented. All study reproductions were implemented based on publicly available information for the studies. After independently attempting to directly reproduce the sampled studies, the team reached out to the corresponding authors of the original papers to discuss how the assumptions regarding study implementation in the reproduction may have differed from the original (author contact protocol available in Supplementary Data 3, author contact files with detailed assumptions Supplementary Data 4). To facilitate discussion, we sent corresponding authors a file

that detailed study reproduction assumptions, a summary of the reproduction protocol and results of the reproduction next to their original study results.

We used transparent processes and software to conduct the reproductions of 150 empirical database studies over the course of 3 years. The reproductions were primarily conducted using the Aetion Evidence Platform® (2021) v4.2 software for real-world data analysis, complemented by SAS 9.4, STATA 14, and Cran R version 3.6.1 when needed. The Aetion platform automatically produced detailed documentation and audit trails of how each study reproduction was implemented, including details of design decisions such as temporal anchors and all code algorithms. The team maintained similar documentation in protocols for studies that were implemented by programmers with SAS, STATA and R. A list of reproduced studies is available in Supplementary Data 5.

### Descriptive measures and statistical analysis

We describe the frequency of reporting of specific types of parameters listed in the consensus document on reporting for RWE study reproducibility, compare characteristics from the reproduction versus the original cohorts, as well as the degree of concordance between measures of association.

The reproducibility of population sample size was measured by dividing the sample size of the original study by the reproduction. For comparative studies, the sample size was determined as the sum of the sample sizes in the compared exposure groups.

We evaluated the reproducibility of baseline characteristics reported in an original manuscript table describing the cohort characteristics. The reproducibility of binary and categorical baseline characteristics of the study population was measured by taking the prevalence of the characteristic reported in the original publication (within each exposure group if there was more than one) and subtracting the prevalence obtained in the reproduction.

The reproducibility of outcome risks and rates was measured by taking the reported outcome risk or rate in the original publication and subtracting the risk or rate obtained in the reproduction. For descriptive studies, the overall risk or rate was reproduced. For comparative studies, the risk or rate was reproduced for each compared group. Rates were converted to reflect events per 100 person-years.

The primary reproducibility metrics for measures of associations of interest were the relative magnitude of the original effect size (e.g., hazard ratio, relative risk, odds ratio) compared to the reproduction effect size (e.g., hazard ratio$_{original}$/hazard ratio$_{reproduction}$) and the Pearson's correlation coefficient between the reproduced and original effect sizes (both unweighted and inverse variance weighted). Other descriptive metrics included the proportion of studies where the absolute difference in the coefficients for the measure of association differed by ≤0.1 or ≤0.2 (e.g., | log(hazard ratio$_{original}$)−log(hazard ratio$_{reproduction}$) |), the proportion of comparative studies where the reproduced measure of association was closer to null than the original, the proportion where the measures were on the same side of null, and the proportion with any overlap in 95% confidence intervals.

We pre-specified measures for study reproducibility mentioned above and show their absolute values in plots or standard descriptive statistics such as means and medians. Arbitrary cutoffs were also used to describe the distribution of these metrics in the manuscript. These cutoffs were not pre-specified. In response to reviewer comments, we computed additional descriptive measures such as the Spearman's rank correlation, the intraclass correlation coefficient, and changes in $p$-value between the original and reproduced studies.

In addition to characterizing how closely measures of association are reproduced, we explored how characteristics of the sampled studies relate to the absolute magnitude of the difference in the coefficients for measures of association in the original study and the reproduction, without multiple-testing adjustment. We analyzed variance in means related to reporting clarity, size of the original effect estimate, data source, funding source, journal type, author experience (proxy measured by first author citation index), calendar year of publication, whether the original study was conducted by investigators in the same research group as the reproduction team, and author responsiveness to questions from the reproduction team.

### Reporting summary

Further information on research design is available in the Nature Research Reporting Summary linked to this article.

## Data availability

Source data are provided with this paper. Source data used to generate tables and figures for this paper are available are provided in a Source Data File using the format requested by the journal. Additionally, the source data are provided in Supplementary Data 6 in a format compatible with the RMarkdown code that has been shared (see code availability below) to reproduce figures and tables. These files have been deposited in the Open Science Framework under accession code https://osf.io/my5gn/. The raw data used to generate the study data are available under restricted access only. Members of the reproduction team were permitted to access the raw data provided by third parties. The raw data are protected and are not publicly available due to data privacy laws and data use agreements. The processed data used to generate tables and figures are available in the Open Science Framework repository (https://doi.org/10.17605/OSF.IO/MY5GN). Our data use agreements for MarketScan, Optum, CPRD and Medicare do not permit us to share source data or data derivatives with individuals and institutions not covered under the agreements. These data sources may be accessed by other investigators through their own data use agreements. The administrative and clinical research databases used in the study reproductions are accessible to other researchers by contacting the data owner/vendors and acquiring data use agreements and/or data licenses. The research data and data derivatives cannot be shared outside of the terms of these agreements. It is our experience that the data vendors we used are very responsive to requests for contracting use of their patient data resources. However, the cost, timeframe, and process for completing the contract for authorized use of these data varies. Contacts and information on how to acquire access to source data: Medicare resdac@umn.edu https://resdac.org/research-identifiable-files-rif-requests Optum Clinformatics connected@optum.com https://www.optum.com/business/solutions/life-sciences/real-world-data/claims-data.html IBM MarketScan https://www.ibm.com/products/marketscan-research-databases/databases CPRD rdg@cprd.com https://www.cprd.com/research-applications The remaining data are available within the Article or from the authors upon request. Source data are provided with this paper.

## Code availability

Analysis code used to generate tables and figures for this paper are available in Supplementary Data 6, located at: https://osf.io/my5gn/, https://doi.org/10.17605/OSF.IO/MY5GN[57].

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

## Author contributions

S.V.W. contributed to conceptualization, methodology, programming and software implementation, validation, formal analysis, investigation, data curation, writing (original draft), writing (reviewing and editing), visualization, supervision, project administration and resource/funding acquisition. S.S. contributed to conceptualization, methodology, programming and software implementation, validation, formal analysis, investigation, data curation, writing (original draft), writing (reviewing and editing), visualization, supervision, project administration and resource/funding acquisition. S.K.S. contributed to programming and software implementation, validation, formal analysis, investigation, data curation, writing (reviewing and editing), supervision, project administration. The co-investigator faculty (J.M.F., J.J.G., K.F.H., E.P.) contributed to methodology, validation, formal analysis, investigation, data curation, writing (reviewing and editing). The reproduction team (Y.J., M.L., M.M., A.P., J.B., L.G.B., K.C., N.G., A.S., E.S., K.S., M.Z., S.D., J.R.R., G.B, J Landon, J Lii, T.T., S.V., E.M.G., L.R.G., M.G., D.L.I., E.P.) contributed to programming and software implementation, validation, formal analysis, investigation, writing (reviewing and editing). The Scientific Advisory Board (S.A., D.B.B., A Bate, A Bourke, J.B., K.A.C., N.C., K.B.F., J.H., L.H., S.H., P.J., K.H.K., A.M., D.M., M.N., B.N., J.S., W.S., L.S., H.T.S., P.T., Y.U., R.W., W.W.) contributed to methodology, investigation, writing (reviewing and editing). We would like to acknowledge the following scientific advisory board members for their contributions: Peter Arlett BSc MBBS MRCP FFPM Pharmacovigilance and Epidemiology Department, European Medicines Agency, Amsterdam, Netherlands Jesse Berlin ScD Johnson & Johnson, New Brunswick, New Jersey Brian Bradbury DSc† Center for Observational Research, Amgen Inc., Thousand Oaks, California Karen Burnett MBA MS Northwestern Memorial Hospital, Chicago Il Troyen Brennan MD CVS Health, Woonsocket, Rhode Island Frank de Vries PharmD PhD Department of Clinical Pharmacy, Maastricht University Medical Center+, Maastricht, Netherlands Lisa Freeman Connecticut Center for Patient Safety, LLC, Fairfield, Connecticut Hans-Georg Eichler MD, MSc Federation of Austrian Social Insurances, Vienna Austria John Ioannidis MD DSc Meta-Research Innovation Center at Stanford, Stanford University, Stanford, California Javier Jimenez MD MPH Real-World Evidence and Clinical Outcomes, Sanofi S.A., Paris, France Christine Laine MD MPH FACP Annals of Internal Medicine, Philadelphia, Pennsylvania Elizabeth Loder MD MPH The BMJ, London, United Kingdom Richard Platt MD MSc Department of Population Medicine, Harvard Medical School, Boston MA Robert W Platt PhD Department of Epidemiology, Biostatistics, and Occupational Health and Department of Pediatrics, McGill University, Montreal, Canada Yoshiaki Uyama PhD Pharmaceuticals and Medical Devices Agency, Tokyo, Japan Deborah Zarin Division of Global Health Equity, Department of Medicine, Brigham and Women's Hospital, Harvard Medical School, Boston, Massachusetts.

## Funding

This project was funded by the non-profit Arnold Ventures, with support from Brigham and Women's Hospital (SVW, SS). Drs. Wang and Schneeweiss were supported by NHLBI RO1HL141505 and NIA R01AG053302 during the conduct of this work.

## Competing interests

S.V.W. received salary support from grants to Brigham and Women's Hospital from Boehringer Ingelheim, Johnson & Johnson and Novartis Pharmaceuticals for unrelated work. She was supported by NHLBI RO1HL141505 and NIA R01AG053302 during the conduct of this work. E.M.G., L.R.G., and M.G. are employees of and have stock options in Aetion, Inc. D.L.I. and E.P. were former employees of Aetion, Inc. during the implementation of this study. Dr. Patorno was supported by the National Institute on Aging (K08AG055670). She is investigator of an investigator-initiated grant to the Brigham and Women's Hospital from Boehringer Ingelheim, not related to the topic of the submitted work. J.J.G. received salary support from grants from Eli Lilly and Company and Novartis Pharmaceuticals Corporation to the Brigham and Women's Hospital and was a consultant to Optum, Inc., all for unrelated work during the conduct of this study. He has since become an employee of Johnson & Johnson. Brian Nosek is Executive Director of the Center for Open Science, a nonprofit technology and culture change organization with a mission to increase openness, integrity, and reproducibility of research. A.B. in an employee of GSK and was at study initiation was an employee of Pfizer. He is a shareholder and hold stock options in GSK and previously held stock and stock options at Pfizer. D.B. is an employee of UCB Pharma. J.M.F. was supported by NHLBI RO1HL141505 during the conduct of this study.

She has since become an employee of Optum Epidemiology. M.P.L. was supported by NHLBI F32 HL149256. J.R.R. was a paid consultant to Aetion during the first year of this study for unrelated work. K.B.F. is supported by a senior salary support award from the *Fonds de recherche du Québec – santé* (Quebec Foundation for Research–Health) and a William Dawson Scholar award from McGill University. David Martin has since become an employee of Moderna. All other authors have no conflict of interest to declare.

## Additional information

## REPEAT Initiative

**Shirley V. Wang** [1,2] ✉, **Sushama Kattinakere Sreedhara**[1], **Sebastian Schneeweiss**[1,2], **Jessica M. Franklin** [1,2], **Joshua J. Gagne**[1,2], **Krista F. Huybrechts**[1,2], **Elisabetta Patorno**[1,2], **Yinzhu Jin**[1], **Moa Lee**[1], **Mufaddal Mahesri**[1], **Ajinkya Pawar**[1], **Julie Barberio**[1], **Lily G. Bessette** [1], **Kristyn Chin**[1], **Nileesa Gautam**[1], **Adrian Santiago Ortiz**[1], **Ellen Sears**[1], **Kristina Stefanini**[1], **Mimi Zakarian**[1], **Sara Dejene**[1], **James R. Rogers**[1], **Gregory Brill**[1], **Joan Landon**[1], **Joyce Lii**[1], **Theodore Tsacogianis**[1], **Seanna Vine**[1], **Elizabeth M. Garry** [3], **Liza R. Gibbs** [3], **Monica Gierada**[3], **Danielle L. Isaman**[3], **Emma Payne** [3], **Sarah Alwardt**[4], **Peter Arlett**[28], **Dorothee B. Bartels**[5], **Andrew Bate** [6], **Jesse Berlin**[29], **Alison Bourke**[7], **Brian Bradbury**[30], **Jeffrey Brown**[8], **Karen Burnett**[31], **Troyen Brennan**[32], **K. Arnold Chan**[9], **Nam-Kyong Choi**[10], **Frank de Vries**[33], **Hans-Georg Eichler**[34], **Kristian B. Filion**[11,12], **Lisa Freeman**[35], **Jesper Hallas**[13], **Laura Happe**[14], **Sean Hennessy**[15], **Páll Jónsson** [16], **John Ioannidis**[36], **Javier Jimenez**[37], **Kristijan H. Kahler**[17], **Christine Laine**[38], **Elizabeth Loder**[39], **Amr Makady**[18], **David Martin**[19], **Michael Nguyen**[19], **Brian Nosek**[20], **Richard Platt**[40], **Robert W. Platt**[41], **John Seeger**[21], **William Shrank**[22], **Liam Smeeth**[23], **Henrik Toft Sørensen** [24], **Peter Tugwell**[25], **Yoshiaki Uyama**[42], **Richard Willke**[26], **Wolfgang Winkelmayer**[27] & **Deborah Zarin**[43]

[3]Aetion, Inc., New York, NY, USA. [4]Avalere Health, Washington, DC, USA. [5]UCB Pharma, Brussels, Germany. [6]GSK, Middlesex, London, UK. [7]IQVIA, London, UK. [8]Department of Population Medicine, Harvard Medical School, Boston, MA, USA. [9]Health Data Research Center, National Taiwan University, Taipei, Taiwan. [10]Department of Health Convergence, Ewha Womans University, Seoul, South Korea. [11]Department of Medicine and Department of Epidemiology, Biostatistics, and Occupational Health, McGill University, Montreal, QC, Canada. [12]Centre for Clinical Epidemiology, Lady Davis Institute, Jewish General Hospital, Montreal, QC, Canada. [13]University of Southern Denmark, Odense, Denmark. [14]Department of Pharmaceutical Outcomes and Policy, University of Florida College of Pharmacy, Gainesville, FL, USA. [15]Perelman School of Medicine, University of Pennsylvania, Philadelphia, PA, USA. [16]National Institute for Health and Care Excellence, London, UK. [17]Evidence & Launch Excellence, Novartis Pharmaceuticals Corporation, East Hanover, NJ, USA. [18]Janssen-Cilag B.V., Breda, Netherlands. [19]Food and Drug Administration, Silver Spring, MA, USA. [20]Center for Open Science, Charlottesville, VA, USA. [21]Optum Epidemiology, Optum, Eden Prairie, MI, USA. [22]Humana, Washington, DC, USA. [23]London School of Hygiene and Tropical Medicine, University of London, London, UK. [24]Aarhus University, Aarhus, Denmark. [25]Journal of Clinical Epidemiology, Ottawa, ON, Canada. [26]The International Society for Health Economics and Outcomes Research, Lawrenceville, NJ, USA. [27]Baylor College of Medicine, Houston, TX, USA. [28]Pharmacovigilance and Epidemiology Department, European Medicines Agency, Amsterdam, Netherlands. [29]Johnson & Johnson, New Brunswick, NJ, USA. [30]Center for Observational Research, Amgen Inc., Thousand Oaks, CA, USA. [31]Northwestern Memorial Hospital, Chicago, IL, USA. [32]CVS Health, Woonsocket, RI, USA. [33]Department of Clinical Pharmacy, Maastricht University Medical Center+, Maastricht, Netherlands. [34]Federation of Austrian Social Insurances, Vienna, Austria. [35]Connecticut Center for Patient Safety, LLC, Fairfield, CT, USA. [36]Meta-Research Innovation Center at Stanford, Stanford University, Stanford, CA, USA. [37]Real-World Evidence and Clinical Outcomes, Sanofi S.A., Paris, France. [38]Annals of Internal Medicine, Philadelphia, PA, USA. [39]The BMJ, London, UK. [40]Department of Population Medicine, Harvard Medical School, Boston, MA, USA. [41]Department of Epidemiology, Biostatistics, and Occupational Health and Department of Pediatrics, McGill University, Montreal, QC, Canada. [42]Pharmaceuticals and Medical Devices Agency, Tokyo, Japan. [43]Division of Global Health Equity, Department of Medicine, Brigham and Women's Hospital, Harvard Medical School, Boston, MA, USA.

