## [Peer Review File · Nature Communications]

Reviewer comments, first round –

Reviewer #1 (Remarks to the Author):

Thank you very much for the opportunity to review this manuscript.

The authors are tackling a very important and relevant research question. They evaluated the transparency (250 articles) and the reproducibility of the analysis (150 reports) of observational studies of routinely collected data (3 administrative databases and 1 primary care database). The authors are completely transparent providing all of the data from the primary analysis and the reproducible analysis for each individual study included.

The manuscript is well written, and the methods are well described and appropriate; nevertheless, I have some comments that could improve the manuscript.

Introduction

1) The introduction provides a very good background for the research questions. The concepts of internal validity, reproducibility and replicability are clearly explained. However, it is unclear to which validity authors are referring in the sentence “this clarity is important for validity assessment”. Please clarify.

Methods section

1) Some methodological decisions should be justified. Particularly

- a) Why search google scholar and not other usual databases such as PubMed or Embase?
- b) Did the authors include only published articles or also preprints?
- c) Any justification for the choice of the sample size?

2) The authors should describe how the 54 item data extraction form was developed. Was it preliminary tested? Secondarily modified?

3) More details on the process used to make assumptions would be helpful: who made the assumptions? What happened in case of discrepancies? Were assumptions discussed and validated with the SAB?

Results section

1) I would strongly recommend not to report specific anecdotes in the main text of the results section. It looks a bit like cherry picking.

2) The authors report the results according to thresholds. For example, they report the % of studies where the reproduction study size was less than half or more than 2 times the original. This presentation is useful to convey the results. However, it would be good to indicate if these thresholds were prespecified or not in the methods section.

3) The authors report univariate analyses for several prespecified covariates. They highlight in the result section that the difference was larger for studies with larger effect sizes. This result is the only statistically significant result. However, this difference could be by chance considering the multiplicity of the tests. Interpretation should be very cautious.

4) The authors summarize the reasons for irreproducibility for the 10 studies with the most extreme differences. It would be more relevant to report the results for the overall sample, particularly the number of studies where no assumptions were necessary as well as the median number of assumptions needed per study. In appendix, the authors list the default assumptions, they should report how often they had to rely on these assumptions. A mapping of all assumptions made with the reasons would also be very helpful to understand why we need to improve transparency.

5) With the results obtained, the authors could make some simple preliminary recommendations on what would be the essential information that should be provided to reduce the need to implement assumptions.

6) Investigators were contacted to validate the assumptions and explore the differences. More details about these results would be interesting, particularly, the number of verified assumptions that were completely validated, partially validated or not validated by authors.

7) I was surprised to see that the original investigator could be in the reproduction team. I would exclude such studies. Could you clarify how often this happened?

Discussion

1) In this research, the authors focused on observational studies where data were accessible upon request. Overall, they did not find strong evidence of bias. However, it would be useful to highlight in the discussion section that in several observational studies, the data are not accessible and such a study is not possible. If we would be able to assess their reproducibility, we may find quite different results.

Abstract: The abstract is one of the most read parts of a manuscript. I do not think the most relevant result of this work is the correlation as we expect a correlation between results. I would encourage the authors to focus on other results, particularly population size, baseline characteristics, outcome risk and rates and effect estimates.

Reviewer #2 (Remarks to the Author):

In this manuscript, the authors attempt to reproduce results from 150 published studies that used one of four databases that the authors were able to get their hands on. They find, for the most part, strong correlation between published primary comparator estimates and reproduced estimates, but the correlation is certainly not perfect. Attempting to reproduce 150 sets of results is a lot of work, and the authors should be congratulated for their efforts. I found the manuscript very interesting and well done, for the most part. I have a few comments / suggestions.

1. I was actually impressed with how closely the results matched; e.g., Figure 5 with correlation of 0.85. (I'd be interested in the rank correlation also.) I was happy to read that the reproduced estimate was closer to the null than the original estimate roughly half (52%) of the time; I kind of expected it to be higher than that. Taken together, these results actually give me a little more confidence in the results of these types of 'real-world evidence' studies. This may go against the authors' narrative, but I believe it is worth highlighting this success a little more. (Of course, this does not mean the associations are correct because of unmeasured confounding, measurement error, etc.). But my impression of this line of research is that it's generally negative and quite critical of the scientific community. Maybe we – and the manuscript authors – should celebrate the successes a little more.

2. I am convinced that reproducibility based on what is written in Methods sections and Supplementary Material (denoted "independent reproducibility" by the authors) is extremely hard to achieve without analysis code. This manuscript is well written and it contains many details necessary for reproducibility. However, I doubt I could reproduce the selection of the exact same 150 studies that they chose for their analysis – despite their detailed description. (I have not attempted – it would obviously be a monumental effort.) I believe the authors are setting the bar too high if they expect perfect reproducibility without analysis code and dated/frozen analysis datasets. For example, I use multiple imputation a lot in my research. There are lots of ways to multiply impute data and different software packages implement multiple imputation in different manners (e.g., the exact implementation of chained equations multiple imputation differs between Stata and R, and it is different within various functions / versions of R). Without the actual analysis code, analysis data, random number seeds, functions used, and versions of the software, perfect reproducibility of results from an analysis that employs multiple imputation is not going to happen. I am not sure it is necessary to put all of these details in the manuscript text, but if authors provide analysis code, then I may be able to achieve perfect reproducibility.

3. Along those lines, the authors seem to downplay the importance of posting analysis code in reproducing findings. The fourth paragraph of the Introduction is all about problems with "computational reproducibility." Their points are valid – but the presentation is skewed. If analysis code was posted for each of those 150 studies, the correlation between original and reproduced studies would certainly be higher than its already high value. How many of the reviewed manuscripts published analysis code? This should be mentioned somewhere in the manuscript and put into Table 1 if it is something greater than zero. If any code was posted, was this code used by the study team in the reproduction efforts? If not, why not? It is true that analysis code can be very long and difficult to read, but it's definitely useful for others to know exactly what was done. I am a big believer in posting analysis code alongside studies.

4. The most startling differences between reproduced and original studies, in my eyes, was due to differences in the study sizes. This may not be surprising if the longitudinal databases are being updated over time – which seems to be the case here for at least some of the databases, and is

common in practice. (As the authors mention, ideally each dataset for each study would be time-stamped and frozen.) Is it possible to present some stratified results based on whether or not the database was updated? Updated data is certainly a problem with reproducing results, but it is hard to blame the original authors if their results cannot be reproduced when the dataset is changing.

5. How much of the difference in point estimates of the comparisons of interest is due to study size (inclusion criteria)? Can the authors provide stratified results so that we can see estimates of similarity (e.g. correlations, differences in effect estimates) among those studies with similar numbers of records included in the original and reproduced results?

6. Did some of the people in this study team participate in the original studies? This seems to be implied by separating studies as to “whether the original study was conducted by investigators in the same research group as the reproduction team.” If that’s the case, it kind of seems like a major design flaw in the current study. Or is it something else. Please explain.

7. The Results section is unreadable without reading the Methods section first. This is more of a critique of the journal style for Nature Communication than a criticism of the article. By relegating Methods (limited to 3000 words) to the end of the article, the journal is downplaying the importance of the Methods – which, as illustrated by the authors, is critical for reproducibility. Journal practices like this – tight word counts, putting the supposedly less-interesting methods section in small print at the end of the article – contribute to the lack of reproducibility.

Minor things:

8. Reference 6 and 40 are repeated. Must be a really good article. :)

9. Figure 2: I assume 0=no and 1=yes. Is this correct? Is this meant to be an example of the type of assumptions that the authors had to make when attempting to reproduce the original analyses? :)
No continuous covariates examined?

10. Why is there a super-imposed normal distribution in Figure 4? What value does this add?

11. I did not understand the difference between these two sentences: “The original and reproduction measures of association were on the same side of the null 82% of the time. The measures of association and 95% confidence intervals were on the same side of the null 61% of the time.”

12. Table 1 shows that when the magnitude of the effect estimate is large the mean of the differences tends to be larger. Could this just be an artifact of extreme estimates often being due to smaller numbers of events and being more variable, so therefore the reproduced estimate is also more variable?

Reviewer #3 (Remarks to the Author):

The authors have studied the reproducibility of real-world evidence studies by trying to reproduce results of 150 of them. This is an important topic, and the study itself represents an enormous

amount of work, for which the authors are commended. Methods are good, and the manuscript itself qualifies for high reproducibility (e.g. the code is given).

1. The introduction rightly insists on RWE to inform the safety or effectiveness of medicinal products. Since those, and in particular effectiveness, are comparative in nature, it remains unclear why descriptive studies were also targeted in the study, and why the sampling of 150 out of 250 was done in order to achieve 80% comparative and 20% descriptive studies. Quantitative results then do not represent the proportion of each in general. Or do they? I would have treated comparative and non-comparative studies separately.
2. I did not find a rationale to target 250 studies to evaluate reporting and 150 for reproduction in the article. Why such numbers? I understand that reproduction of results is an incredible task, so 150 is perfectly fine with me, but evaluating reporting could have been done for more eligible studies. And at the end only 8 were excluded (but this was unpredictable I guess).
3. Results show that authors were able to reproduce results for more than half studies. I would have thought as +/- 10% in effect estimate as indicating a good reproducibility if the dataset and analysis code had to be both reconstructed, and even a 20% difference would be quite acceptable. I'd say that the way the abstract is worded gives the impression of a half-empty glass, instead of half-full glass, which would have been my own interpretation. Of course, I understand that the authors may have a different opinion. But what is more important is the cases where the differences were larger (here, the top 10 differences were taken). The discussion of those cases is interesting, but this makes less than 10% of all studies. So, again, one may rather conclude that for some studies there were issues, but this was only a minority, and in the majority of cases, results were quite reproducible.
4. As a follow-up comment, the conclusion of the abstract "Greater methodological transparency aligned with new guidance will improve reproducibility and validity assessment, thus facilitating evidence-based decision-making" does not seem to completely match the results. Again, reproducibility was not bad, and there are cases where the main issue was not methodological transparency (see the point below). So do the results really show that greater methodological transparency will improve reproducibility? At least more cautious wording is necessary.
5. On the reasons for study lack of reproducibility, the figure 6 is really uninformative in a scientific paper. One may expect at least in how many cases the different reasons were found. Or another quantitative analysis.
6. Changes in data due to dramatic changes to the databases themselves (e.g. anecdote 3) should not be part of reproducibility, if we follow the authors' definition. Discrepancies due to such issues are therefore quite challenging when we want to interpret the reproducibility of the results linking it with how authors of original papers report their methods (in those examples, even with perfect description and analysis code available, results may not be reproduced). The manuscript is quite clear on describing this issue, but I wondered if this should not be quantified more clearly. There is an important topic here: either this amounts to the lack of reproducibility of RWE analyses, or it is a slightly different issue, that could be factored out in the paper. I leave it to the authors, but this should be addressed more clearly.
7. The discussion is balanced on the fact that the conclusions apply to studies using those well-established and curated, and exclusively US and UK, databases, but the abstract is less clear on that limitation.
8. The authors also acknowledge that there is no single perfect metrics to characterise reproducibility. However the correlation coefficient has been described as poorly suited to assess reproducibility for long. Often, intraclass correlation coefficients have been considered as more appropriate (although they do not lack limitations either).
9. In terms of data presentation, why not use Bland-Altman plots to better display the agreement

between the original and reproduced estimates? They could supplement the observed vs. reproduced plots and better emphasize how many results were outside the 0.1, 0.2, 0.3 ... range of difference.

10. The use of cut-offs on both the relative and log-difference scales (e.g. ratio of HR between 0.9 and 1.1, and then absolute difference of the difference of $\log HR > 0.2$) adds unnecessary complexity (and this difference is absolutely not clear in the abstract). Why not use a single scale: e.g. $RHR < 0.8$ or > 1.25 to indicate larger differences, or difference of $\log HR$ between -0.1 and 0.1 to indicate small difference?

11. In terms of metrics, the authors report the cases where confidence intervals did not overlap, point estimates and confidence intervals on the same side of the null. Even if we do not trust statistical significance so much, it could be interesting to look at studies where the conclusions would be markedly changed in the re-analysis, e.g. when a p-value < 0.01 or < 0.001 (which many reader would regard as indicating fair evidence of a non-null effect) would become > 0.05 , or the reverse. I agree that all those thresholds are arbitrary, but this may highlight situations where the conclusions of the study would be overturned, avoiding the changes from a p-value of 0.045 to 0.055 in the re-analysis (which may also raise the question of p-hacking, but it's a different topic).

12. In the table 1, how were the ANOVA p-values obtained? I would have expected that results would be meta-analysed by subgroup, for instance with an inverse variance weighting estimator, and that between subgroup differences would be tested, still by a meta-analytic approach. Is it what "ANOVA p-value" means here?

13. Were there also cases where the sample size, sample characteristics or incidence rates were quite different, but not the effect estimates? Would such results indicate stability of the original findings?

Minor comments

1. How reasonable is the sentence "it is critically important to ensure that there is clear communication of all study parameters regardless of whether shared data and analysis code are available"? The term "study parameters" is not clearly defined, but it may comprise so many elements (e.g. exact model formulation for imputing missing data, random generator and seed used whenever they would be used to impute data or select a match among others, etc.) How come could this be fully reported without giving access to the code? So either we should ask for the code (or make it mandatory), or we should accept some level of precision cannot be attained.

2. 80% of 150 should make 118. But I assume that no more than 118 comparative studies could be found in the 250 selected ones. But in the 258?

3. In the table 1, are mean differences in HR, RR or OR really reported? Not mean differences in their logarithms? This would add a third way of expressing results, with less relevance than the ratio or difference of logs.

Point by point response to reviewer comments provided in blue.

REVIEWER COMMENTS

Reviewer #1 (Remarks to the Author):

Thank you very much for the opportunity to review this manuscript.

The authors are tackling a very important and relevant research question. They evaluated the transparency (250 articles) and the reproducibility of the analysis (150 reports) of observational studies of routinely collected data (3 administrative databases and 1 primary care database). The authors are completely transparent providing all of the data from the primary analysis and the reproducible analysis for each individual study included.

Thank you for the recognition of the importance and relevance of this work.

The manuscript is well written, and the methods are well described and appropriate; nevertheless, I have some comments that could improve the manuscript.

Introduction

1) The introduction provides a very good background for the research questions. The concepts of internal validity, reproducibility and replicability are clearly explained. However, it is unclear to which validity authors are referring in the sentence "this clarity is important for validity assessment". Please clarify.

We have revised the section to read as follows:

"...This clarity is necessary for assessment of potential sources of bias (e.g. confounding, misclassification, selection bias)..."

Methods section

1) Some methodological decisions should be justified. Particularly

We have added sentences to explain our rationale as below.

a) Why search google scholar and not other usual databases such as PubMed or Embase?

"We chose to use Google Scholar because we were interested in studies that were conducted using the specific databases for which we had access. This information is often not provided in the title and abstract of manuscripts. Unlike PubMed and Web of Science, Google Scholar searches the full text when available."
(<https://libguides.lib.msu.edu/pubmedvsgoogle scholar>)

b) Did the authors include only published articles or also preprints?

“Studies were excluded if the search result referred to a poster, a conference abstract, pre-print, or members of the review team were unable to access a PDF for a full manuscript.”

c) Any justification for the choice of the sample size?

“The target sample size for this descriptive characterization of reproducibility was chosen based on estimates for the largest feasible number of studies that could be evaluated within the project time frame.”

2) The authors should describe how the 54 item data extraction form was developed. Was it preliminary tested? Secondarily modified?

The 54-item data extraction form used the elements recently identified by a joint task force between the International Society of Pharmacoepidemiology and International Society of Pharmacoconomics and Outcomes Research as being necessary to report for reproducibility and validity assessment. We have included a citation to that work and the following sentence in the methods section.

“Evaluation of the clarity in reporting of 250 study implementations was based on a standardized extraction form covering 54 items corresponding to elements of a consensus document cataloguing study parameters necessary for reproducibility and validity assessment.”

3) More details on the process used to make assumptions would be helpful: who made the assumptions? What happened in case of discrepancies? Were assumptions discussed and validated with the SAB?

The SAB were not involved in the choices of assumptions to make for specific studies. We have added sentences to provide more details on the process to make assumptions.

“A set of default assumptions was created for study parameters that were frequently ambiguous from publications (**Data S2**). If there was no relevant default assumption and a study parameter was unclear, the assumption was based on the best interpretation of the reproduction team given the context provided in the paper. Each paper had at least 2 research staff and one faculty member involved in the reproduction. The research staff proposed assumptions to make and reviewed these assumptions with the faculty member that they were teamed with. If the team was not able to come to agreement, another faculty member was involved. After discussion, the assumptions made during reproduction were documented.”

Results section

1) I would strongly recommend not to report specific anecdotes in the main text of the results section. It looks a bit like cherry picking.

We disagree with that interpretation and note that this was not a concern brought up by other reviewers. In our results section, we start by reporting the overall descriptive statistics

in the whole sample. We then provide what is clearly labeled as an anecdote. While the anecdote may be non-representative, by illustrating one of the more extreme results as a case study, it helps the reader to understand what sorts of issues can be driving the observed variability in the summary result. This is a format that was also taken in a recently published paper on the reproducibility of cancer biology studies. (Errington et al 2021, "Investigating the replicability of preclinical cancer biology", <https://elifesciences.org/articles/71601>)

2) The authors report the results according to thresholds. For example, they report the % of studies where the reproduction study size was less than half or more than 2 times the original. This presentation is useful to convey the results. However, it would be good to indicate if these thresholds were prespecified or not in the methods section.

We have added the following sentences:

"We pre-specified measures for study reproducibility mentioned above and show their absolute values in plots or standard descriptive statistics such as means and medians. Arbitrary cutoffs were also used to describe the distribution of these metrics in the manuscript. These cutoffs were not pre-specified. In response to reviewer comments, we computed additional descriptive measures such as the Spearman's rank correlation, the intraclass correlation coefficient, and changes in p-value between the original and reproduced studies."

3) The authors report univariate analyses for several prespecified covariates. They highlight in the result section that the difference was larger for studies with larger effect sizes. This result is the only statistically significant result. However, this difference could be by chance considering the multiplicity of the tests. Interpretation should be very cautious.

We have revised our interpretation to be more cautious.

"The differences between the original and reproduced effect sizes were larger on average for studies with larger published effect sizes than studies with smaller published effect sizes ($p < 0.01$). However, this could be an artifact driven by studies with few outcomes that produced highly variable but large effect sizes, or, given the multiple factors explored, a chance finding."

4) The authors summarize the reasons for irreproducibility for the 10 studies with the most extreme differences. It would be more relevant to report the results for the overall sample, particularly the number of studies where no assumptions were necessary as well as the median number of assumptions needed per study. In appendix, the authors list the default assumptions, they should report how often they had to rely on these assumptions. A mapping of all assumptions made with the reasons would also be very helpful to understand why we need to improve transparency.

A detailed mapping of the assumptions for each study that was reproduced was provided for each study in what was previously supplemental data S6, now labeled supplemental data S4. The frequency with which we had to make assumptions were additionally summarized in

supplemental tables 1 and 2. These two tables describe the frequency of original authors providing information about a series of specific study parameters (e.g. inclusion-exclusion, exposure, outcome, covariates, follow up, etc.). If the authors did not provide information on their decisions for a specific parameter, then we made an assumption. We have added to supplemental table 1, a crude tally of the distribution of the number of categories in which assumptions were needed.

In our tally, we collapsed the assumptions into 6 binary yes/no buckets corresponding to assumptions pertaining to index date, inclusion-exclusion criteria, exposure, outcome, covariates, and follow up such that any number of assumptions about any number of inclusion-exclusion criteria was counted as “yes”. Thus, the maximum number in the tally was 6 for comparative studies and 5 for descriptive studies (which did not focus on a specific exposure). This tally is a crude measure because there were weak assumptions (e.g. assuming that covariates were captured using any diagnosis position in any care setting) and stronger assumptions (e.g. the clinical code list used to identify the outcome + the care setting + the diagnosis position). These are not expected to have equal effect on reproducibility.

We have added the following text:

“Out of 6 categories for comparative studies (each category combining multiple study parameters used to define index date, inclusion-exclusion criteria, exposure, outcome, follow up, covariates), the median and interquartile range for the number of categories where the reproduction team made assumptions was 4 [3, 5]. Out of 5 categories for descriptive studies (where no specific exposure was studied) the median and interquartile range for the number of categories where the reproduction team made assumptions was 3 [2, 4] (**SM Table 1**). Only 3 out of 250 studies did not require an assumption in any of these categories.”

5) With the results obtained, the authors could make some simple preliminary recommendations on what would be the essential information that should be provided to reduce the need to implement assumptions.

Our take home was that we needed at least all the information in the 54-item extraction sheet based on a consensus document. There was no one or smaller set of items that stood out as the drivers of irreproducibility.

“No specific study parameter stood out as being strongly associated with reproducibility in our univariate descriptive analyses. We noted that even studies that were closely reproduced often required considerable discussion within the team, sometimes with many assumptions about the original implementation decisions due to ambiguity in the methods description. The prevalence of RWE studies that could not be closely reproduced speaks to the need for higher levels of transparency and expectations when communicating critical details of data sources, study design, analysis, and implementation in protocols, publications and reports. Examining the details of these studies suggested that the divergence was often multi-factorial. Because we observed that the reproduction of most studies required the team to make assumptions on at least one key parameter from a consensus document that outlined elements needed for reproducibility and validity assessment (the basis of our 54-item extraction form), aiming to meet at least this level of detail in reporting on future studies would be a substantial step forward.”

6) Investigators were contacted to validate the assumptions and explore the differences. More details about these results would be interesting, particularly, the number of verified assumptions that were completely validated, partially validated or not validated by authors.

We have added the following details:

“Out of 150 corresponding authors of reproduced studies, 53% responded, 33% did not respond after 3 attempts at contact, and 13% of the e-mails to corresponding authors were undeliverable. Among 81 responders, 32% (n = 26) provided helpful clarification or stated that they were comfortable with our assumptions, 32% (n = 26) provided somewhat helpful feedback that did not address all of the assumptions, 27% (n = 22) provided responses that did not address the assumptions, and 12% (n = 6) declined to discuss their study, citing lack of interest or time.”

7) I was surprised to see that the original investigator could be in the reproduction team. I would exclude such studies. Could you clarify how often this happened?

We have clarified with the following statement:

“Eight of the selected studies were published by members of the same research department as members of the reproduction team. In these cases, the paper was assigned to research staff and faculty who were not involved with the original publication.”

Discussion

1) In this research, the authors focused on observational studies where data were accessible upon request. Overall, they did not find strong evidence of bias. However, it would be useful to highlight in the discussion section that in several observational studies, the data are not accessible and such a study is not possible. If we would be able to assess their reproducibility, we may find quite different results.

To clarify, our study was focused on evaluating the reproducibility of RWD studies, not of bias. We were in some cases able to reproduce studies that made flawed choices that resulted in bias because the original investigators were clear in their reporting of their design and analysis decisions. As noted in paragraph 5 of the discussion:

“Other studies have described the prevalence of issues that bias RWE studies and we focused on reproducibility without systematic evaluation of the appropriateness of design or analytic choices. While close independent reproducibility of a study is a marker for completeness of reporting on study design and analysis methods, **reproducibility is not itself an indicator of high study validity**. Indeed, clear reporting in studies with methodological problems enabled our team to closely reproduce results that suffered from the same limitations.”

We have added a sentence to the discussion noting that the reproducibility of studies for whom the research data cannot be accessed by independent parties may differ than what we observed.

“Further, the reproduction effort is focused on US and UK data sources frequently used in research and the generalizability may be limited to well-established and curated research databases that are accessible to independent researchers.”

Abstract: The abstract is one of the most read parts of a manuscript. I do not think the most relevant result of this work is the correlation as we expect a correlation between results. I would encourage the authors to focus on other results, particularly population size, baseline characteristics, outcome risk and rates and effect estimates.

We would like to include more information on results in the abstract, as suggested by the reviewer. However, the journal has a tight 150-word limit for the abstract. While we would expect a correlation in effect sizes due to the nature of the reproduction, a correlation in population size, baseline characteristics, outcome risks and rates is also expected because we are using the same data sources and trying to apply the same methods. We reported as much as we could in the abstract given the limited word count.

Reviewer #2 (Remarks to the Author):

In this manuscript, the authors attempt to reproduce results from 150 published studies that used one of four databases that the authors were able to get their hands on. They find, for the most part, strong correlation between published primary comparator estimates and reproduced estimates, but the correlation is certainly not perfect. Attempting to reproduce 150 sets of results is a lot of work, and the authors should be congratulated for their efforts. I found the manuscript very interesting and well done, for the most part. I have a few comments / suggestions.

We thank the reviewer for the positive comments.

1. I was actually impressed with how closely the results matched; e.g., Figure 5 with correlation of 0.85. (I'd be interested in the rank correlation also.) I was happy to read that the reproduced estimate was closer to the null than the original estimate roughly half (52%) of the time; I kind of expected it to be higher than that. Taken together, these results actually give me a little more confidence in the results of these types of 'real-world evidence' studies. This may go against the authors' narrative, but I believe it is worth highlighting this success a little more. (Of course, this does not mean the associations are correct because of unmeasured confounding, measurement error, etc.). But my impression of this line of research is that it's generally negative and quite critical of the scientific community. Maybe we – and the manuscript authors – should celebrate the successes a little more.

The reproduction team was comprised of researchers whose field is focused on doing RWD analyses. We were cautious of being perceived as shading things too positively. Therefore, we aimed for a neutral interpretation of the observed results that described the current observed state and highlighted where things could be improved (because there is always

room for improvement). We aimed for language that was neither celebrating nor castigating/casting judgement on the scientific community.

We have added Spearman's rank correlation in the results. In response to another reviewer, we have also added the intraclass correlation coefficient.

"Different statistics indicated strong correlation between the original and reproduced measures of associations (Fig. 5). The unweighted and inverse variance weighted Pearson's correlation coefficient between the original and reproduced measures of association were 0.85 and 0.79, respectively. The unweighted and inverse variance weighted Spearman's Rank Correlation were 0.82 and 0.87. The intraclass correlation coefficient and 95% Confidence Interval was 0.85 (0.79, 0.89)."

2. I am convinced that reproducibility based on what is written in Methods sections and Supplementary Material (denoted "independent reproducibility" by the authors) is extremely hard to achieve without analysis code. This manuscript is well written and it contains many details necessary for reproducibility. However, I doubt I could reproduce the selection of the exact same 150 studies that they chose for their analysis – despite their detailed description. (I have not attempted – it would obviously be a monumental effort.) I believe the authors are setting the bar too high if they expect perfect reproducibility without analysis code and dated/frozen analysis datasets. For example, I use multiple imputation a lot in my research. There are lots of ways to multiply impute data and different software packages implement multiple imputation in different manners (e.g., the exact implementation of chained equations multiple imputation differs between Stata and R, and it is different within various functions / versions of R). Without the actual analysis code, analysis data, random number seeds, functions used, and versions of the software, perfect reproducibility of results from an analysis that employs multiple imputation is not going to happen. I am not sure it is necessary to put all of these details in the manuscript text, but if authors provide analysis code, then I may be able to achieve perfect reproducibility.

We agree with the reviewer that perfect reproducibility is achievable with the availability of source longitudinal patient data and all code used to create study populations/cohorts from the source data as well as all code used to analyze the created datasets. However, that refers to computational reproducibility. We are not focused on perfect, computational reproducibility and are not expecting it in this study. As we describe in the introduction (paragraphs 4 and 5), computational reproducibility is different from independent reproducibility.

We have added/ revised the following text:

"While computational reproducibility will allow reproduction of the same exact result to the nth decimal place, independent reproducibility focuses on effective communication of critical design and analytic choices (with the potential to affect the validity, relevance or interpretation of results). This clarity is necessary for assessment of potential sources of bias (e.g. confounding, misclassification, selection bias) as well as to facilitate replication of findings with data that is stored in different data model. Thus, independent reproducibility is critically important because it reflects the clarity of communication of key study design and

analytic parameters, regardless of whether shared data and analysis code are available.”

3. Along those lines, the authors seem to downplay the importance of posting analysis code in reproducing findings. The fourth paragraph of the Introduction is all about problems with “computational reproducibility.” Their points are valid – but the presentation is skewed. If analysis code was posted for each of those 150 studies, the correlation between original and reproduced studies would certainly be higher than its already high value. How many of the reviewed manuscripts published analysis code? This should be mentioned somewhere in the manuscript and put into Table 1 if it is something greater than zero. If any code was posted, was this code used by the study team in the reproduction efforts? If not, why not? It is true that analysis code can be very long and difficult to read, but it’s definitely useful for others to know exactly what was done. I am a big believer in posting analysis code alongside studies.

Of the selected studies, 6 provided a reference to a programming macro or open-source code and 4 provided a procedure or command. When analytic code was referenced and available to us, we used it. However, the references to code did not provide information on all the settings used for the package/procedure.

We have added a sentence to that effect:

“Analytic code in the form of macros, other open-source code, or specific procedures was referenced in 7% of reproduced studies. However, the exact software version or selected options that were used to run the code for these studies were only partially provided.”

See responses to the above point about our focus on independent reproducibility rather than computational reproducibility and why.

We state up front that sharing code is useful. However, our point is that sharing code is not a substitute for true and broadly understood transparency.

Shared code can be of limited value when the data cannot be shared or if the code is specific to a particular data model. For example, publicly available code used to create analytic cohorts from healthcare databases stored in the Sentinel, PCORnet or OMOP common data models is unusable for data stored in other models. Without access to the source data stored in a particular data model, sharing of large packages of interweaving scripts used to create study populations from longitudinal source data can provide an artificial sense of transparency. Only a very small subset of people will be able to use the posted code, much less understand what it is doing. We are encouraging researchers to provide language and visualizations, as well as code lists and phenotypes to effectively communicate what they did and allow shared understanding. Shared understanding of methodological choices can be measured by independent reproducibility.

4. The most startling differences between reproduced and original studies, in my eyes, was due to differences in the study sizes. This may not be surprising if the longitudinal databases are being updated over time – which seems to be the case here for at least some of the databases, and is common in practice. (As the authors mention, ideally each dataset for each

study would be time-stamped and frozen.) Is it possible to present some stratified results based on whether or not the database was updated? Updated data is certainly a problem with reproducing results, but it is hard to blame the original authors if their results cannot be reproduced when the dataset is changing.

There is no blame being assigned. The reality of what we live with is that source data is being updated over time. While the differences in sample sizes could also have been due to the assumptions made about how inclusion-exclusion criteria were defined, data shifts over time and its impact on independent reproducibility points to the need to think about how to better document and report on data versions and updates.

Given the information that was available from the original publications, we cannot know for certain whether the source databases were updated. However, the lag between the time the original investigators accessed the data source and when the reproduction team accessed the data would suggest that each of the data sources involved for this study was likely updated in ways that could have small or large effects.

5. How much of the difference in point estimates of the comparisons of interest is due to study size (inclusion criteria)? Can the authors provide stratified results so that we can see estimates of similarity (e.g. correlations, differences in effect estimates) among those studies with similar numbers of records included in the original and reproduced results?

We note that the differences in estimates could have come from a mix of sources, including different study size or population characteristics (who is included), different data versioning, or differences in outcome ascertainment. Nevertheless, we have added stratified results using arbitrary cut-offs for small, medium, or large differences in original and reproduced study sizes (Table 3, SM 2B, SM Fig 3). The following sentences have been added.

“The distribution of differences was similar for descriptive versus comparative study types as well as for studies where there were small, medium or large differences sample size between the original and the reproduction (SM Fig. 2).”

“The distribution in the relative magnitude of measures of association was similar regardless of whether there were small, medium or large differences in the original and reproduced study sample sizes. (SM Fig. 3)”

6. Did some of the people in this study team participate in the original studies? This seems to be implied by separating studies as to “whether the original study was conducted by investigators in the same research group as the reproduction team.” If that’s the case, it kind of seems like a major design flaw in the current study. Or is it something else. Please explain.

We have clarified with the following statement:

“Eight of the selected studies were published by members of the same research department as members of the reproduction team. In these cases, the paper was assigned to research staff and faculty who were not involved with the original publication.”

7. The Results section is unreadable without reading the Methods section first. This is more of a critique of the journal style for Nature Communication than a criticism of the article. By relegating Methods (limited to 3000 words) to the end of the article, the journal is downplaying the importance of the Methods – which, as illustrated by the authors, is critical for reproducibility. Journal practices like this – tight word counts, putting the supposedly less-interesting methods section in small print at the end of the article – contribute to the lack of reproducibility.

We refer this comment to the editorial team at Nature Communication.

Minor things:

8. Reference 6 and 40 are repeated. Must be a really good article. :)

It is a seminal article... :)
Nevertheless, it does not need to be repeated. Thank you for pointing out this error.

9. Figure 2: I assume 0=no and 1=yes. Is this correct? Is this meant to be an example of the type of assumptions that the authors had to make when attempting to reproduce the original analyses? :) No continuous covariates examined?

Thank you for pointing out the lack of clarity in figure 2. We have updated the key.

There were very few continuous covariates reported for each study. Furthermore, the standard deviation was often not reported. Because the scale of the continuous covariates varied quite widely, absolute differences between the original and reproduction were hard to summarize. Without standard deviations we could not compare standardized differences. Therefore, we focused on binary/categorical covariates.

10. Why is there a super-imposed normal distribution in Figure 4? What value does this add?

We have removed the super-imposed normal distribution in figure 4.

11. I did not understand the difference between these two sentences: "The original and reproduction measures of association were on the same side of the null 82% of the time. The measures of association and 95% confidence intervals were on the same side of the null 61% of the time."

The former refers to only point estimates being on the same side of null. The latter refers to both point estimates AND confidence intervals being on the same side of null. The latter has a smaller proportion because in some cases the point estimates were on the same side of null, but the confidence intervals for the original study included null and the reproduction did not, or vice versa. We have revised the language as follows:

“The point estimates of the original and reproduction measures of association were on the same side of null 82% of the time. The point estimates for the measures of association and the 95% confidence intervals were on the same side of null 61% of the time.”

12. Table 1 shows that when the magnitude of the effect estimate is large the mean of the differences tends to be larger. Could this just be an artifact of extreme estimates often being due to smaller numbers of events and being more variable, so therefore the reproduced estimate is also more variable?

Yes, this is possible. We have incorporated this point into the interpretation.

“The difference between the original and reproduced effect sizes were larger on average for studies with larger published effect sizes than studies with smaller published effect sizes ($p < 0.01$). However, this could be an artifact driven by studies with few outcomes that produced highly variable but large effect sizes, or, given the multiple factors explored, a chance finding.”

Reviewer #3 (Remarks to the Author):

The authors have studied the reproducibility of real-world evidence studies by trying to reproduce results of 150 of them. This is an important topic, and the study itself represents an enormous amount of work, for which the authors are commended. Methods are good, and the manuscript itself qualifies for high reproducibility (e.g. the code is given).

We thank the reviewer for their positive assessment.

1. The introduction rightly insists on RWE to inform the safety or effectiveness of medicinal products. Since those, and in particular effectiveness, are comparative in nature, it remains unclear why descriptive studies were also targeted in the study, and why the sampling of 150 out of 250 was done in order to achieve 80% comparative and 20% descriptive studies. Quantitative results then do not represent the proportion of each in general. Or do they? I would have treated comparative and non-comparative studies separately.

Epidemiology is about the study of the distribution and determinants of disease. We were interested in evaluating the reproducibility of studies describing the incidence/prevalence of disease as well as causal inference studies on medical interventions. Therefore, we included both descriptive and comparative studies. Because we expected that the reproducibility of effect sizes for causal inference studies would be of great importance, we chose to oversample the comparative studies. Because only the comparative studies had measures of association, any results on effect size reproducibility reflect comparative studies only.

We had some discussion on whether to produce tables describing clarity of reporting and reproducibility of cohort characteristics for descriptive and comparative cohorts combined or separately and ended up choosing to combine them because the process of creating descriptive and comparative cohorts from longitudinal source data is the same and we did

not observe meaningful differences in distribution. We have added results stratified by descriptive versus comparative study type to the supplement (Supplemental Table 1, Supplemental Figure 2 A)

2. I did not find a rationale to target 250 studies to evaluate reporting and 150 for reproduction in the article. Why such numbers? I understand that reproduction of results is an incredible task, so 150 is perfectly fine with me, but evaluating reporting could have been done for more eligible studies. And at the end only 8 were excluded (but this was unpredictable I guess).

We have included the following rationale:

“The target sample size for this descriptive characterization of reproducibility was chosen based on estimates for the largest feasible number of studies that could be evaluated within the project time frame.”

3. Results show that authors were able to reproduce results for more than half studies. I would have thought as +/- 10% in effect estimate as indicating a good reproducibility if the dataset and analysis code had to be both reconstructed, and even a 20% difference would be quite acceptable. I'd say that the way the abstract is worded gives the impression of a half-empty glass, instead of half-full glass, which would have been my own interpretation. Of course, I understand that the authors may have a different opinion. But what is more important is the cases where the differences were larger (here, the top 10 differences were taken). The discussion of those cases is interesting, but this makes less than 10% of all studies. So, again, one may rather conclude that for some studies there were issues, but this was only a minority, and in the majority of cases, results were quite reproducible.

As we stated in our response to another reviewer, we were aiming for a neutral interpretation of the observed results that described the current observed state and highlighted where things could be improved (because there is always room for improvement). It was not our intent to give the impression of a half-empty glass.

Our conclusion was that for some studies there were issues, but for the majority the results were relatively closely reproducible. We have revised the last sentences of the abstract to make that more clear.

“Studies that generate real-world evidence on the effects of medical products through analysis of digital data collected in clinical practice provide key insights for regulators, payers, and other healthcare decision-makers. Ensuring reproducibility of such findings is fundamental to effective evidence-based decision-making. We sought to reproduce results for 150 studies published in peer-reviewed journals using the same research databases as original investigators and evaluated the completeness of reporting for 250. Original and reproduction effect sizes were positively correlated (Pearson's 0.85), a strong but imperfect relationship. The median and IQR for the relative magnitude of effect (e.g. hazard ratio_{original} / hazard ratio_{reproduction}) was 1.0 [0.9, 1.1], range [0.3, 2.1]. While the majority of results were closely reproduced, a subset were not. The latter could be explained by incomplete reporting and updates to research data. Greater methodological transparency aligned with new

guidance may further improve reproducibility and validity assessment, thus facilitating evidence-based decision-making.”

A similar edit was made in the discussion:

“While our reproduced estimates were relatively close to the original estimates from the majority of studies, there was a subset of findings that the study team was unable to closely recreate, even though we applied the reported methods to the same healthcare data sources as the original authors.”

4. As a follow-up comment, the conclusion of the abstract “Greater methodological transparency aligned with new guidance will improve reproducibility and validity assessment, thus facilitating evidence-based decision-making” does not seem to completely match the results. Again, reproducibility was not bad, and there are cases where the main issue was not methodological transparency (see the point below). So do the results really show that greater methodological transparency will improve reproducibility? At least more cautious wording is necessary.

See revised abstract pasted above with more cautious wording. “..may further improve”

5. On the reasons for study lack of reproducibility, the figure 6 is really uninformative in a scientific paper. One may expect at least in how many cases the different reasons were found. Or another quantitative analysis.

We have removed figure 6. Tables 1 and 2 in the supplemental appendix report how often the reproduction team was not able to identify the original investigator’s decision on specific study parameters.

6. Changes in data due to dramatic changes to the databases themselves (e.g. anecdote 3) should not be part of reproducibility, if we follow the authors’ definition. Discrepancies due to such issues are therefore quite challenging when we want to interpret the reproducibility of the results linking it with how authors of original papers report their methods (in those examples, even with perfect description and analysis code available, results may not be reproduced). The manuscript is quite clear on describing this issue, but I wondered if this should not be quantified more clearly. There is an important topic here: either this amounts to the lack of reproducibility of RWE analyses, or it is a slightly different issue, that could be factored out in the paper. I leave it to the authors, but this should be addressed more clearly.

We have added the following sentences:

“For RWE studies, part of transparency on methods to enable independent reproducibility is communicating sufficient detail to allow identification of the relevant data or data version. While many updates to data may have negligible effect on research findings, if there are substantial changes to the contents of the data resources, this may affect the assessment of the fitness of the data resource for the research question.”

7. The discussion is balanced on the fact that the conclusions apply to studies using those well-established and curated, and exclusively US and UK, databases, but the abstract is less clear on that limitation.

Unfortunately, with a tight 150-word limit in the abstract, we could not elaborate on this limitation in the abstract and instead expand on it in the discussion.

8. The authors also acknowledge that there is no single perfect metrics to characterise reproducibility. However the correlation coefficient has been described as poorly suited to assess reproducibility for long. Often, intraclass correlation coefficients have been considered as more appropriate (although they do not lack limitations either).

We recognize that the correlation coefficient is imperfect and others have limitations as well, therefore we report multiple metrics, each with their own limitations to try and characterize reproducibility. We have added the Spearman rank correlation (in response to another reviewer comment) and intraclass correlation coefficient (in response to your comment).

“Different statistics indicated strong correlation between the original and reproduced measures of associations (Fig. 5). The unweighted and inverse variance weighted Pearson’s correlation coefficient between the original and reproduced measures of association were 0.85 and 0.79, respectively. The unweighted and inverse variance weighted Spearman’s Rank Correlation were 0.82 and 0.87. The intraclass correlation coefficient and 95% Confidence Interval was 0.85 (0.79, 0.89).”

9. In terms of data presentation, why not use Bland-Altman plots to better display the agreement between the original and reproduced estimates? They could supplement the observed vs. reproduced plots and better emphasize how many results were outside the 0.1, 0.2, 0.3 ... range of difference.

We find calibration plots to be more intuitively understood. However, we have produced Bland-Altman plots and included them in the manuscript or appendix. We have also added the following text:

“The funnel shape of Bland-Altman plots show larger reproduction differences when the averaged risk or rate from the original and reproduction was larger (Supplemental Figure 4A).”

“A Bland-Altman plot did not show a clear relationship between effect size and reproducibility (Fig. 6).”

10. The use of cut-offs on both the relative and log-difference scales (e.g. ratio of HR between 0.9 and 1.1, and then absolute difference of the difference of $\log HR > 0.2$) adds unnecessary complexity (and this difference is absolutely not clear in the abstract). Why not use a single scale: e.g. $RHR < 0.8$ or > 1.25 to indicate larger differences, or difference of $\log HR$ between -0.1 and 0.1 to indicate small difference?

We have revised the abstract so that there are only cut-offs on one scale reported (median, IQR, range). We are able to provide more description in the actual manuscript to clarify the different cutoffs that were used to characterize reproducibility in different ways.

11. In terms of metrics, the authors report the cases where confidence intervals did not overlap, point estimates and confidence intervals on the same side of the null. Even if we do not trust statistical significance so much, it could be interesting to look at studies where the conclusions would be markedly changed in the re-analysis, e.g. when a p-value <0.01 or <0.001 (which many reader would regard as indicating fair evidence of a non-null effect) would become >0.05 , or the reverse. I agree that all those thresholds are arbitrary, but this may highlight situations where the conclusions of the study would be overturned, avoiding the changes from a p-value of 0.045 to 0.055 in the re-analysis (which may also raise the question of p-hacking, but it's a different topic).

We report the difference in p-values for studies that obtained point estimates on the same side of null and the proportion that moved from a <0.01 to >0.05 or vice versa.

“When studies had point estimates on the same side of null, the median and interquartile range for the difference in p-values between the original and reproduced studies was 0.00 [-0.05, 0.00]”

“In 16% of studies, the p-value for the study reproduction was less than or equal to 0.01 (a stringent threshold for rejecting the null) while the p-value for the original study was greater than 0.05 (an arbitrary threshold for failing to reject the null hypothesis) or vice versa

12. In the table 1, how were the ANOVA p-values obtained? I would have expected that results would be meta-analysed by subgroup, for instance with an inverse variance weighting estimator, and that between subgroup differences would be tested, still by a meta-analytic approach. Is it what “ANOVA p-value” means here?

We did a simple 1-way ANOVA to test for differences in means.

13. Were there also cases where the sample size, sample characteristics or incidence rates were quite different, but not the effect estimates? Would such results indicate stability of the original findings?

Yes, there were cases where we were astonished that the effect sizes were as close as they were given the degree of assumptions that the reproduction team had to make and the variability in reproducibility of cohort characteristics. This could indicate stability of findings or coincidence/chance.

Minor comments

1. How reasonable is the sentence “it is critically important to ensure that there is clear communication of all study parameters regardless of whether shared data and analysis code

are available”? The term “study parameters” is not clearly defined, but it may comprise so many elements (e.g. exact model formulation for imputing missing data, random generator and seed used whenever they would be used to impute data or select a match among others, etc.) How come could this be fully reported without giving access to the code? So either we should ask for the code (or make it mandatory), or we should accept some level of precision cannot be attained.

We have revised the section as follows:

“While computational reproducibility will allow reproduction of the same exact result to the n th decimal place, independent reproducibility focuses on effective communication of critical design and analytic choices (with the potential to affect the validity, relevance or interpretation of results). This clarity is necessary for assessment of potential sources of bias (e.g. confounding, misclassification, selection bias) as well as to facilitate replication of findings with data that is stored in a different data model. Thus, independent reproducibility is critically important because it reflects the clarity of communication of key study design and analytic parameters, regardless of whether shared data and analysis code are available.”

2. 80% of 150 should make 118. But I assume that no more than 118 comparative studies could be found in the 250 selected ones. But in the 258?

Although we were targeting 120 comparative studies, 2 descriptive studies in the reproduction pipeline were incorrectly labeled as comparative. We have added a sentence to that effect.

“The target was for 80% (120) of the sampled studies for reproduction to focus on a comparative safety or effectiveness question, however due to an error in categorization early in the reproduction pipeline, only 118 comparative studies were reproduced and an extra 2 descriptive studies were included.”

3. In the table 1, are mean differences in HR, RR or OR really reported? Not mean differences in their logarithms? This would add a third way of expressing results, with less relevance than the ratio or difference of logs.

The title of the table states that it is reporting on the mean difference of the coefficient for the HR, RR and OR (therefore the difference of logs).

Reviewer comments, second round –

Reviewer #1 (Remarks to the Author):

The reviewer adequately answered and took into account my comments. I have no other request

Reviewer #2 (Remarks to the Author):

I feel that the authors were fairly responsive to my earlier comments, and after re-reading the manuscript, I still found it interesting and insightful.

Many of my comments were similar to those made by another reviewer. In particular, we both felt that the authors' interpretation of the reproducibility of their findings was 'a glass half-empty' sort of interpretation. On reading the authors' response and re-reading the manuscript, I still think the authors tend to highlight the negative more than the positive, but I do not think their interpretation is excessively negative, so I am OK with this.

The biggest study limitation in my eyes is the difficulty teasing out exactly why the lack of reproducibility -- as some of it may be due to poor reporting in original manuscripts (which researchers can control) and some of it may be due to evolving databases (which researchers really cannot control, but I guess can better report when the database was extracted).

A secondary limitation is the authors' choice to include papers written by researchers from their same department. I guess ensuring that researchers who attempted the reproduction are different from those who did the original paper partially mitigates this concern, but it still is not ideal. I am a lot more likely to reply to an email from a colleague down the hallway than from a stranger I do not know.

On this second read, it was not clear to me whether the final reproduced numbers were those obtained before or after contacting the original authors for clarification of assumptions. I guess either way is OK. If the final reproduced numbers are those before contacting the original authors then it is a better assessment of reproducibility based on what is written in the manuscript ("independent reproducibility"?). However, if the final reproduced numbers are those after contacting the original authors then perhaps this is a more pragmatic assessment of reproducibility -- although is it really still "independent"? In either case, this should be clarified.

Reviewer #3 (Remarks to the Author):

I thank the authors for their answers and clarification.

- Still a very minor point remaining pertains to wording. In the legends of table 1 and figure 5, "logarithms of HR, RR or RR" would be clearer than "coefficient for HR ...".
- Otherwise, the references 53, 55 and 56 should be checked: as they are cited (e.g. "I. W. Health. "), it is virtually impossible to find those if we do not know what they are in advance.

RESPONSE TO REVIEWER COMMENTS

Reviewer #1 (Remarks to the Author):

The reviewer adequately answered and took into account my comments. I have no other request
Thank you, we are glad that we were able to address the reviewer's comments.

Reviewer #2 (Remarks to the Author):

I feel that the authors were fairly responsive to my earlier comments, and after re-reading the manuscript, I still found it interesting and insightful.

Thank you, we are glad that we were able to address the reviewer's comments.

Many of my comments were similar to those made by another reviewer. In particular, we both felt that the authors' interpretation of the reproducibility of their findings was 'a glass half-empty' sort of interpretation. On reading the authors' response and re-reading the manuscript, I still think the authors tend to highlight the negative more than the positive, but I do not think their interpretation is excessively negative, so I am OK with this.

We have made further revisions to try and re-iterate that this is not a glass half empty situation. By and large we were able to do pretty well in terms of reproduction, but there is always room for improvement, and this project provided some insights on where we could improve.

“There is always room for improvement and this project has provided insights on how to improve transparency and reproducibility. With coordinated effort from key stakeholders, standards for clear and reproducible RWE studies can be set higher to facilitate efficient evaluation of validity and effective, evidence-based decision-making.”

“While the majority of results are closely reproduced, a subset are not. The latter can be explained by incomplete reporting and updated data.”

“In 150 RWE studies we showed that the resulting effect sizes were strongly correlated between the original publications and study reproductions in the same data sources and developed insights on where there was room for improvement.”

The biggest study limitation in my eyes is the difficulty teasing out exactly why the lack of reproducibility – as some of it may be due to poor reporting in original manuscripts (which researchers can control) and some of it may be due to evolving databases (which researchers really cannot control, but I guess can better report when the database was extracted).

We agree that this is challenging because reproducibility is multi-factorial and have so noted in the manuscript discussion.

“No specific study parameter stood out as being strongly associated with reproducibility in our univariate descriptive analyses, highlighting the fact that independent reproducibility is multi-factorial.”

A secondary limitation is the authors choice to include papers written by researchers from their same department. I guess ensuring that researchers who attempted the reproduction are different from those who did the original paper partially mitigates this concern, but it still is not ideal. I am a lot more likely to reply to an email from a colleague down the hallway than from a stranger I do not know.

On this second read, it was not clear to me whether the final reproduced numbers were those obtained before or after contacting the original authors for clarification of assumptions. I guess either way is OK. If the final reproduced numbers are those before contacting the original authors then it is a better assessment of reproducibility based on what is written in the manuscript ("independent reproducibility"?). However, if the final reproduced numbers are those after contacting the original authors then perhaps this is a more pragmatic assessment of reproducibility – although is it really still "independent"? In either case, this should be clarified.

We have clarified in the manuscript that the reproduced numbers are independent reproductions. The numbers were produced solely based on publicly reported information, before reaching out to corresponding authors.

“The results that are reported reflect independent attempts to reproduce the original publication results, prior to reaching out to the original investigative team.”

Reviewer #3 (Remarks to the Author):

I thank the authors for their answers and clarification.

- Still a very minor point remaining pertains to wording. In the legends of table 1 and figure 5, “logarithms of HR, RR or RR” would be clearer than “coefficient for HR ...”.

We have revised the legends of Table 1 and Figure 5 as recommended by the reviewer.

- Otherwise, the references 53, 55 and 56 should be checked: as they are cited (e.g. “I. W. Health. “), it is virtually impossible to find those if we do not know what they are in advance.

Thank you for noting the reference formatting errors. We have corrected references 53, 55, and 56.